# An integrated multi-layer 3D-fabrication of PDA/RGD coated graphene loaded PCL nanoscaffold for peripheral nerve restoration

Yun Qian[1,2,3], Xiaotian Zhao[1], Qixin Han[4], Wei Chen[2], Hui Li[5] & Weien Yuan[1]

As a conductive nanomaterial, graphene has huge potentials in nerve function restoration by promoting electrical signal transduction and metabolic activities with unique topological properties. Polydopamine (PDA) and arginylglycylaspartic acid (RGD) can improve cell adhesion in tissue engineering. Here we report an integrated 3D printing and layer-by-layer casting (LBLC) method in multi-layered porous scaffold fabrication. The scaffold is composed of single-layered graphene (SG) or multi-layered graphene (MG) and polycaprolactone (PCL). The electrically conductive 3D graphene scaffold can significantly improve neural expression both in vitro and in vivo. It promotes successful axonal regrowth and remyelination after peripheral nerve injury. These findings implicate that graphene-based nanotechnology have great potentials in peripheral nerve restoration in preclinical and clinical application.

[1] School of Pharmacy, Shanghai Jiao Tong University, 800 Dongchuan Road, Shanghai 200240, China. [2] Department of Orthopedics, Shanghai Jiao Tong University Affiliated Sixth People's Hospital, 600 Yishan Road, Shanghai 200233, China. [3] Shanghai Sixth People's Hospital East Campus, Shanghai University of Medicine and Health, Shanghai 201306, China. [4] Renji Hospital, School of Medicine, Shanghai Jiao Tong University, Shanghai 200135, China. [5] School of Medicine, University of California, 1450 Third St., San Francisco, CA 94158, USA. Correspondence and requests for materials should be addressed to W.Y. (email: yuanweien@sjtu.edu.cn)

The nerve guidance conduit can connect two nerve ends, guide axonal regeneration and offer a neurotrophic environment to Schwann cell aggregation and proliferation[1]. Two major factors are involved in fabricating an excellent nerve conduit. The first factor is material selection. Synthetic nerve scaffolds are gaining increasing popularity and exhibit better long-term performance than natural products[2]. Among these materials, polycaprolactone (PCL) is a kind of biocompatible[3], biodegradable[4], and nontoxic material[5]. It has many successful applications in the nerve regeneration. PCL can connect severely damaged nerve stumps with excellent mechanical properties. In addition, its appropriate flexibility and rigidity are beneficial to Schwann cells and endothelial cells migration as well as long-term support for regenerated nerves in the conduit[6]. However, autografts still exhibit better outcomes than nerve conduits. Recently, an emerging conductive material, graphene has received wide attention. Graphene exhibits two-dimensional (2D) structure formed by carbon $sp^2$ hybridization. It is composed of different layers of sheets[7]. It resembles graphite properties when the layer number reaches ten or above[8]. The honeycomb planar structure is formed by one carbon atom and its adjacent three atoms. Therefore, graphene family have different electrically conductive and mechanical properties[9]. It was previously reported that graphene improved cell bioelectricity between biocompatible scaffold and cellular membrane because it had a strong $\pi$ bond and a large surface area. The biological activities of nerve stem cells could be significantly upregulated by three-dimensional (3D) fabrication of graphene nanocomposite channels[10]. In addition, it displays a higher electron transfer velocity than present electronic materials such as carbon nanotubes[7]. Therefore, graphene provides significant mechanical and biochemical cues for successful nerve restoration.

The second factor is surface modification and scaffold fabrication. Although graphene possesses high electric conductivity, the strong $\pi$ bond poses potential damages to the cell membrane integrity[11]. For graphene and its derivatives, appropriate surface modification not only increases their affinity to cell membrane but also decreases cytotoxicity, which requires proper surface coating[12]. Therefore, graphene conduit modification is important to ideal biocompatibility. Nucleic acid[13], polypeptides[14], and cell seeding[15] are commonly used in surface coating. Polydopamine (PDA) can improve cell adhesion and serve as substrates of bioactive scaffolds[16], because it exhibits high hydrophilicity, enduring anti-erosion ability and excellent biochemical properties[17]. Apart from PDA, arginylglycylaspartic acid (RGD) consists of three amino acids including L-arginine, glycine, and L-aspartic acid. It can improve cell adhesion and therefore is widely used in the tissue engineering[18]. Although RGD can improve adhesion, it easily vanishes in physiological atmosphere[19]. Controlled release design can prevent it from dissolving in the surroundings quickly[20]. The joint effect of PDA and RGD has been investigated in bone regeneration[21]. However, the combined application of PDA and RGD has not been reported in peripheral nerve restoration.

In this study, we innovatively invented an integrated layer-by-layer casting (LBLC) method for nerve conduit fabrication. Compared with traditional electrospinning fabrication, our method successfully overcomes many problems like failure in quality control, weak mechanical strength, random gaps between nanofibers, and most importantly uneven drug delivery distribution[22]. Therefore, we fabricate a multilayered 3D graphene conduit. It improves axonal regeneration and remyelination after physical nerve injury. It indicates that graphene-based nanomaterials may contribute to successful peripheral nerve restoration in preclinical and clinical applications.

## Results

**Fabrication and characterization of graphene nanoscaffolds.** In this study, an integrated LBLC method was used to fabricate the 3D porous graphene conduit (Fig. 1). A 3D printer was composed of a rolling tube and a sprayer. A tubular mode with evenly distributed microneedles was placed on a roller. The nozzle above this rolling tube directly sprayed different solutions on the rolling tube. The tube was rolling at a constant speed and added different layers to form a tubular structure. The first and also the inner-most part was RGD and PDA mixed layer. And then graphene/PCL dichloromethane solution was sprayed on the rolling tube and crosslinked with PDA/RGD layer. Another single-layered or multi-layered graphene and PCL mixed solution was sprayed again after previous layers were solidified. Finally, RGD and PDA mixed solution was sprayed again to form the outer-most layer. The inner-most and outer-most layers of RGD and PDA were beneficial to cell adhesion and proliferation. The graphene/PCL double layers could intensify the tubular structure and allow certain stiffness for long-term in vivo study.

A reasonable pore size is vital to successful nerve conduit fabrication because it allows free entrance of water and oxygen into the lumen. Either too big or too small pore size interferes with ideal peripheral nerve restoration[23]. In this study, we used microneedles to create multiple aligned macropores with 50 μm in diameter in the conduit. We removed the microneedles and the rolling tube mold after the conduit was solidified.

We characterized the morphology of the porous 3D graphene conduit by optical imaging and scanning electron microscopy (SEM) (VEGA3 TESCAN, Fig. 2). The nanoporous and multi-layered 3D structure was shown at different magnifications. From Raman spectra results, the 2D peak is a single and sharp one in single-layered graphene sheet. However, it was relatively low in multi-layered graphene sheet. The rescaled image showed a higher intensity for 2D peak in single-layered graphene. The 2D peak was relatively 1/2 the height of G peak in multi-layered graphene, while the 2D peak was two times as high as G peak in single-layered graphene. The position of 2D band from single-layered graphene sheet was around 2676 cm$^{-1}$, however it was 2682 cm$^{-1}$ for multi-layered graphene sheet (Supplementary Fig. 1). The Raman spectrum helped us distinguish the different carbon structures of single-layered and multi-layered graphene because the reduction in layers resulted in different electronic dispersions. To test the mechanical property of the 3D conduit, we measured the elastic modulus and found that the average value for single-layered graphene/PCL conduit was 68.74 MPa, in contrast with 58.63 MPa for multi-layered graphene/PCL conduit. The mechanical test indicated that the porous 3D graphene conduit could hold the structure and allow nerve regrowth by offering ideal flexibility and rigidity. We also evaluated the electric conductivity for different scaffolds. The single-layered graphene/PCL conduit displayed a high conductivity of $8.92 \times 10^{-3}$ S cm$^{-1}$. The electric conductivity of multi-layered graphene/PCL conduit was $6.37 \times 10^{-3}$ S cm$^{-1}$. This was consistent with previous research that the electric conductivity decreased with more layers of graphene[24]. In addition, it displayed relatively good electric conductivity like some conductive materials. For instance, Song and colleagues focused on excellent conductive material polypyrrole (PPY). The electric conductivity of PPY based conduit was $6.72 \times 10^{-5}$ S cm$^{-1}$[25]. Our multi-layered macroporous nerve conduit could allow exchanges of nutrients and oxygen via excellent permeability, strong mechanical support, appropriate biodegradation rate, and flexibility for complete nerve regrowth.

After nerve conduit fabrication, we seeded Schwann cells on the nanoscaffolds and evaluated the neural expression. Furthermore, this 3D conduit was also implanted in a long-range sciatic nerve defect model in Sprague Dawley (SD) rats.

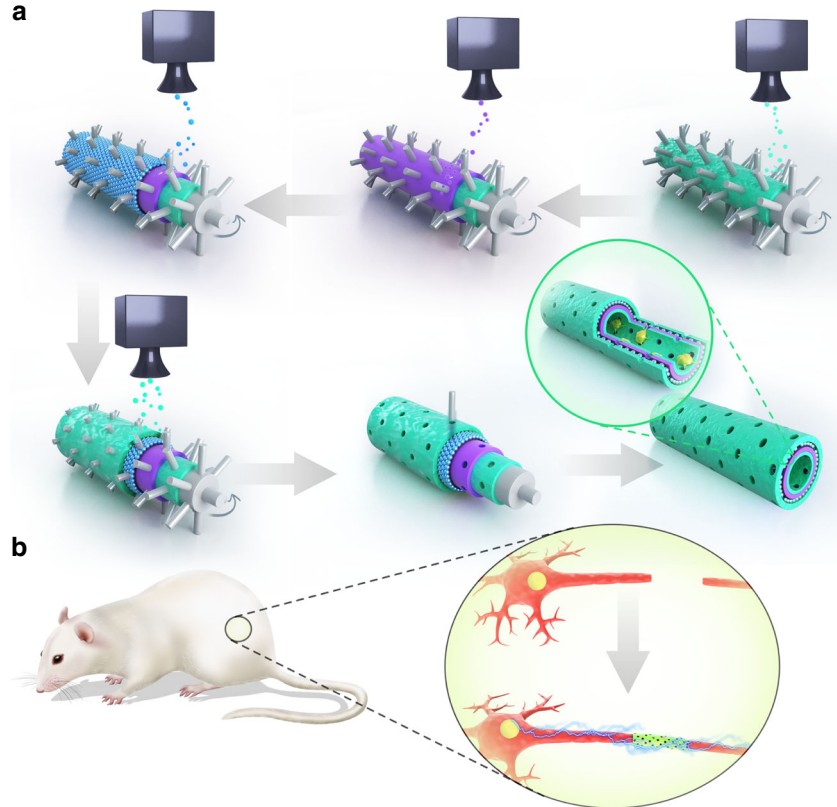

**Fig. 1** Schematic illustration of graphene nerve conduit fabrication with LBLC method. **a** The inner-most and outer-most green layers are PDA/RGD mixed layers. The purple layer is single-layered or multi-layered graphene and PCL mixed layer. The blue layer is a repetition of the graphene and PCL mixed layer. **b** An illustration of the single-layered or multi-layered graphene/PCL nerve conduit in a sciatic nerve defect model in the SD rats

## Cell proliferation and attachment on graphene nanoscaffolds.

To verify rat Schwann cell (RSC) viability on the nanoscaffolds, different graphene/PCL was designed to determine the concentration dependent cytotoxicity on Schwann cells by cell counting kit 8 (CCK8) assay, including 0.1%, 0.5%, 1%, 2%, and 4% single-layered and multi-layered graphene in PCL. CCK8 assay showed that 1% single-layered and multi-layered graphene displayed lower cytotoxicity than 2% and 4% single-layered and multi-layered graphene. In addition, cells were more proliferative in 1% than 0.5% and 0.1% single-layered and multi-layered graphene. Therefore, we chose 1% graphene and further evaluated its effects in peripheral nerve regeneration. This was consistent with previous research. Park and colleagues evaluated the toxicity of graphene nanoparticle both in vitro and in vivo. They found 0.5% graphene was the maximal concentration for a relatively high cell viability and low systematic organ damages. In contrast, 1% and 2% graphene nanoplatelets exerted a negative influence on normal cell and tissue function[26]. In this study, we modified single-layered and multi-layered graphene with PDA and RGD in a controlled release way and confirmed 1% graphene nanoparticle was suitable for optimal cell biocompatibility.

To verify that graphene scaffolds could support cell proliferation and attachment, we seeded Schwann cells on the different scaffolds for 1, 3, 5, and 7 days respectively and examined their proliferative state by CCK8 assay. After 1, 3, and 5 days respectively, SCs on PDA/RGD-SG/PCL, PDA/RGD-MG/PCL, PDA/RGD-PCL, and PCL nanoscaffolds were similarly proliferative as tissue culture plate (TCP) (analysis of variance (ANOVA), $p > 0.05$, Fig. 3). At day 7, the outcomes of PDA/RGD-SG/PCL and PDA/RGD-MG/PCL were significantly better than other groups, indicating the positive role of PDA and RGD-modified graphene in cell proliferation. In addition, PDA/RGD-

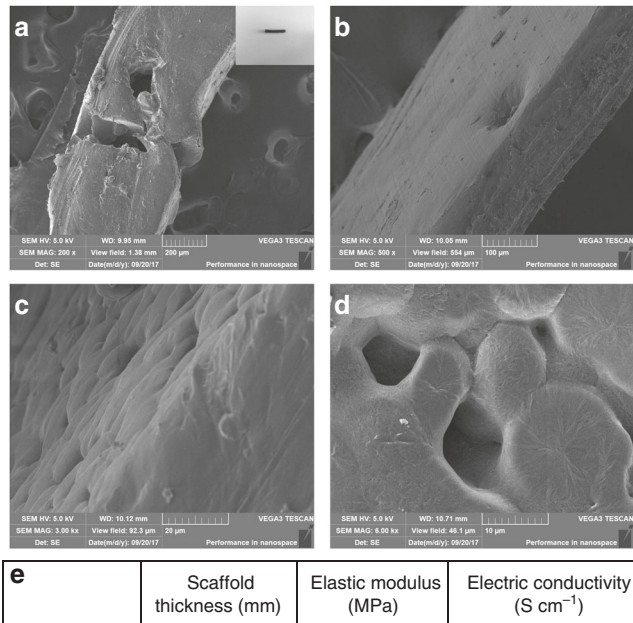

**Fig. 2** Characterization of graphene nerve conduit. **a–d** SEM images for evaluation of the nanoporous and multi-layered 3D structure in graphene-based nanomaterials. **e** Thickness, elastic modulus, and electric conductivity of PDA/RGD-SG/PCL and PDA/RGD-MG/PCL scaffolds (evaluation of both materials was repeated for five times)

|  | Scaffold thickness (mm) | Elastic modulus (MPa) | Electric conductivity (S cm⁻¹) |
|---|---|---|---|
| PDA/RGD-SG/PCL | 0.48 | 68.74 | $8.92\times10^{-3}$ |
| PDA/RGD-MG/PCL | 0.46 | 58.63 | $6.37\times10^{-3}$ |

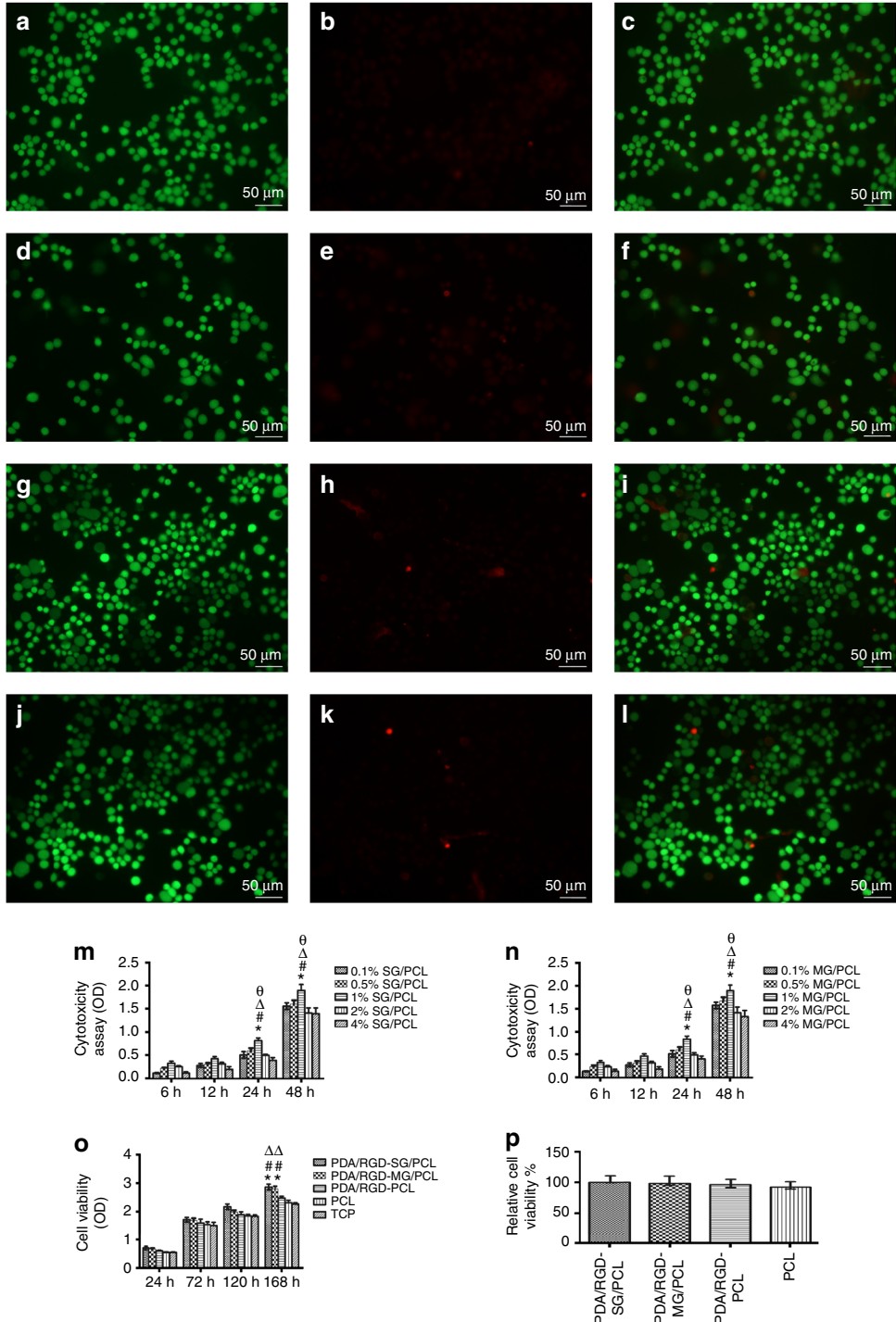

**Fig. 3** Cell viability assay of LIVE/DEAD cell staining and CCK8. **a–c** Live/dead/merge pictures for PDA/RGD-SG/PCL. **d–f** Live/dead/merge pictures for PDA/RGD-MG/PCL. **g–i** Live/dead/merge pictures for PDA/RGD-PCL. **j–l** Live/dead/merge pictures for PCL. The scale bar is 50 μm. **m** Cytotoxicity assay for 0.1%, 0.5%, 1%, 2%, and 4% SG/PCL at different time points. **n** Cytotoxicity assay for 0.1%, 0.5%, 1%, 2%, and 4% SG/PCL at different time points. *$p$ < 0.05 compared with 0.1% SG(MG)/PCL; #$p$ < 0.05 compared with 0.5% SG(MG)/PCL; $^{\Delta}p$ < 0.05 compared with 2% SG(MG)/PCL. $^{\Phi}p$ < 0.05 compared with 4% SG(MG)/PCL. **o** CCK8 assay for five groups. **p** Relative cell viability by live and dead staining. All data are displayed as mean ± standard deviation. *$p$ < 0.05 compared with PDA/RGD-PCL; #$p$ < 0.05 compared with PCL; $^{\Delta}p$ < 0.05 compared with TCP (the statistical test is ANOVA)

SG/PCL nanoscaffolds could improve the greatest extent of cell proliferation among three PDA/RGD coated scaffolds.

LIVE/DEAD cell kit was used for cell viability analysis. Figure 3 exhibited live and dead cells which were stained with Calcein AM and Ethidium homodimer-1. The results of various nanoscaffolds did not display a notable difference and all of them showed optimal cell viability after one-day co-culture.

We performed immunofluorescence and western blotting (WB) to further evaluate cell proliferation and attachment on the different scaffolds. N-cadherin and vinculin are adhesion-associated proteins and can activate chemical signaling via extracellular matrix (ECM). The expression of vinculin and N-cadherin on PDA/RGD-SG/PCL and PDA/RGD-MG/PCL nanoscaffolds was significantly increased compared with that on

PDA/RGD-PCL and PCL nanoscaffolds (Fig. 4). This indicated that graphene could also prominently contribute to SCs adhesion just like PDA and RGD. In addition, we evaluated the proliferative ability of SCs on the different nanoscaffolds by Ki67 and Brdu. The expression of Brdu and Ki67 on PDA/RGD-PCL and PCL nanoscaffolds was relatively lower than that on PDA/RGD-SG/PCL and PDA/RGD-MG/PCL nanoscaffolds. Ki67 showed a higher expression on PDA/RGD-SG/PCL nanoscaffolds (Fig. 4). Immunofluorescent staining of Ki67 was shown in Fig. 5.

SEM was performed to evaluate cell morphology after SCs were seeded on the different nanoscaffolds for 3 days. Cells were evenly distributed on the entire nanoscaffolds and almost covered all the fields. Protuberances of most cells were extended on PDA/RGD-SG/PCL and PDA/RGD-MG/PCL nanoscaffolds. By phalloidin staining, the cell density was also increased in PDA/RGD-SG/PCL and PDA/RGD-MG/PCL nanoscaffolds (Fig. 5). These results indicated that the graphene nanoscaffolds were able to improve cell attachment.

**Neural expression on graphene nanoscaffolds**. The glial fibrillary acidic protein (GFAP), Class III β-tubulin (Tuj1), and S100 were involved in the immunofluorescence assay to validate single-layered graphene/PCL and multi-layered graphene/PCL scaffolds could promote neural expression (Fig. 6). Tuj1 can distinguish neurons from glial cells. GFAP is expressed in many nerve cell types in the central and peripheral nerve system. The relative expression level of GFAP on PDA/RGD-SG/PCL was respectively 3.8-fold, 5.4-fold, and 7.5-fold greater than that of cells cultured on PDA/RGD-MG/PCL, PDA/RGD-PCL, and PCL nanoscaffolds. Furthermore, the relative expression level of Tuj1 on PDA/RGD-SG/PCL nanoscaffold was respectively 1.2-fold, 2.3-fold, and 2.8-fold greater than that of cells cultured on PDA/RGD-MG/PCL, PDA/RGD-PCL, and PCL nanoscaffolds. For further validation, WB was also shown in Fig. 4. Neurotrophic factors have profound implications in peripheral nerve regeneration, such as nerve growth factor (NGF), brain-derived neurotrophic factor (BDNF), glial cell line-derived neurotrophic factor (GDNF), and ciliary neurotrophic factor (CNTF). The expression of NGF, BDNF, GDNF, and CNTF on PDA/RGD-SG/PCL nanoscaffolds was apparently higher than that on other nanoscaffolds (Supplementary Fig. 2). Conclusively speaking, PDA/RGD-MG/PCL and PDA/RGD-SG/PCL had huge potentials in promoting neural expression and differentiation.

**Functional improvement of graphene conduits**. The in vitro study confirmed the potential effects of graphene-based nanoscaffolds in promoting cell growth and neural expression. Further in vivo evaluation would help us identify the long-term performance of graphene-based conduit in peripheral nerve restoration. Ninety SD rats were allocated into six groups randomly and equally, including Schwann cell-loaded PDA/RGD-SG/PCL, Schwann cell-loaded PDA/RGD-MG/PCL, PDA/RGD-SG/PCL, PDA/RGD-MG/PCL, PDA/RGD-PCL, and autograft groups. Each group was evaluated at 6 weeks, 12 weeks, and 18 weeks after surgery.

We did not observe severe complications like delay of wound healing, ulcer, infection at 6, 12, and 18 weeks after surgery. No nerve conduits degraded at 18 weeks after surgery. All regenerated nerves were observed by optical imaging (Supplementary Fig. 3).

To evaluate functional recovery in all experimental rats, walking track analysis was performed according to a previous research[27]. The sciatic function index (SFI) results were displayed (Supplementary Fig. 4). At 6 and 12 weeks after surgery, the

recovery of sciatic nerves was significantly faster in Schwann cell-loaded PDA/RGD-SG/PCL and Schwann cell-loaded PDA/RGD-MG/PCL nerve conduits than the remaining scaffolds (ANOVA, $p < 0.05$), but it was not as good as the autograft group (ANOVA, $p < 0.05$). At 18 weeks after surgery, the Schwann cell-loaded PDA/RGD-SG/PCL and Schwann cell-loaded PDA/RGD-MG/PCL nerve conduits showed similar results compared with the autograft group (ANOVA, $p > 0.05$). In addition, we also evaluated extensor postural thrust, which was also an important indicator for motor performance. The Schwann cell-loaded single-layered and multi-layered graphene/PCL conduit could improve the motor functions better than non-cell loading conduit groups and PCL conduit group at 6, 12, and 18 weeks post operatively (ANOVA, $p < 0.05$). Meanwhile, they were as excellent as the autograft group at 18 weeks after surgery (ANOVA, $p > 0.05$, Supplementary Fig. 3). It also showed the beneficial effects of graphene-based nerve conduit and cell loading in the sciatic nerve functional recovery.

Gastrocnemius muscle recovery can also indicate nerve function because it is dominated by sciatic nerves. We weighed the gastrocnemius muscle and calculated the average weight. At 6 and 12 weeks, there was a statistical difference among Schwann cell-loaded PDA/RGD-SG/PCL, Schwann cell-loaded PDA/RGD-MG/PCL nerve conduits, and other conduits (ANOVA, $p < 0.05$). Without cell loading, muscles from PDA/RGD-SG/PCL and PDA/RGD-MG/PCL groups also showed significant higher weight than PDA/RGD-PCL group (ANOVA, $p < 0.05$). Cell-loaded conduit groups showed similar results in comparison with the autograft group at 18 weeks after surgery (ANOVA, $p > 0.05$, Supplementary Fig. 4). It indicated that single- and multi-layered graphene could reverse muscle atrophy and help nerve recovery to a certain extent, which was further improved by Schwann cell loading.

Apart from locomotor function recovery, we also evaluated sensory functional recovery in all groups. Significant increase in response time was observed in Schwann cell-loaded PDA/RGD-SG/PCL, Schwann cell-loaded PDA/RGD-MG/PCL, PDA/RGD-SG/PCL, PDA/RGD-MG/PCL, and PDA/RGD PCL groups compared with the autograft group at 6 and 12 weeks. In contrast, the values of cell-loaded conduit groups were close to the autograft group at 18 weeks, and were significantly better than those of the other groups (ANOVA, $p < 0.05$, Supplementary Fig. 3). This indicated a successful sensory recovery after graphene conduit implantation and cell loading therapy.

**Electrophysiological improvement of graphene conduits**. SD rats were subjected to electrophysiological analysis to evaluate electrophysiological performance. We performed electrophysiological analysis at 6 and 12 weeks post operatively. The nerve conducting velocity (NCV) of the Schwann cell-loaded PDA/RGD-SG/PCL (14.8 m s$^{-1}$, 21.2 m s$^{-1}$) and Schwann cell-loaded PDA/RGD-MG/PCL nerve conduits (13.4 m s$^{-1}$, 20.7 m s$^{-1}$) was notably higher than that of the remaining conduit groups (PDA/RGD-SG/PCL: 11.1 m s$^{-1}$, 17.1 m s$^{-1}$; PDA/RGD-MG/PCL: 10.9 m s$^{-1}$, 16.2 m s$^{-1}$; PDA/RGD-PCL: 9.4 m s$^{-1}$, 13.7 m s$^{-1}$, ANOVA, $p < 0.05$). However, it was significantly lower than the autograft group (17.2 m s$^{-1}$, 25.2 m s$^{-1}$, ANOVA, $p < 0.05$). We achieved similar results at 18 weeks post operatively. The NCV of Schwann cell-loaded PDA/RGD-SG/PCL and Schwann cell-loaded PDA/RGD-MG/PCL groups was significantly higher than that of PDA/RGD-SG/PCL, PDA/RGD-MG/PCL, and PDA/RGD-PCL groups. However, it showed no significant differences compared with the autograft group (ANOVA, $p > 0.05$). We evaluated the distal compound motor action potential (DCMAP) and achieved similar results at 6, 12, and 18 weeks after surgery (Supplementary Fig. 4).

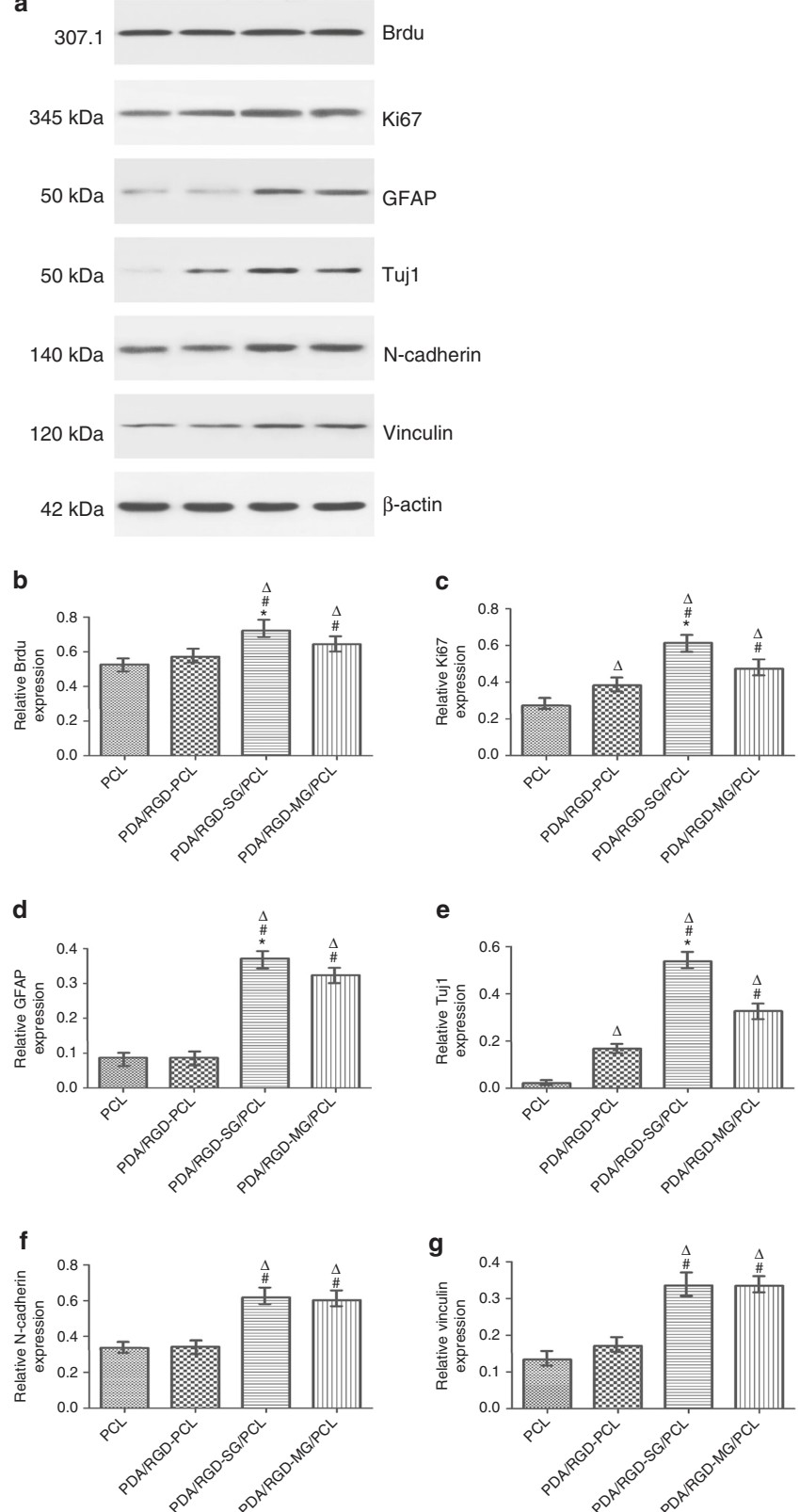

**Fig. 4** Gene expression compared between nanoscaffolds. **a** WB assay of Ki67, Brdu, GFAP, Tuj1, N-cadherin, and vinculin. **b**–**g** Their relative expression from SC seeded PDA/RGD-SG/PCL, PDA/RGD-MG/PCL, PDA/RGD-PCL, and PCL nanoscaffolds. All data are displayed as mean ± standard deviation. *$p$ < 0.05 compared with PDA/RGD-MG/PCL; #$p$ < 0.05 compared with PDA/RGD-PCL; Δ$p$ < 0.05 compared with PCL (the statistical test is ANOVA)

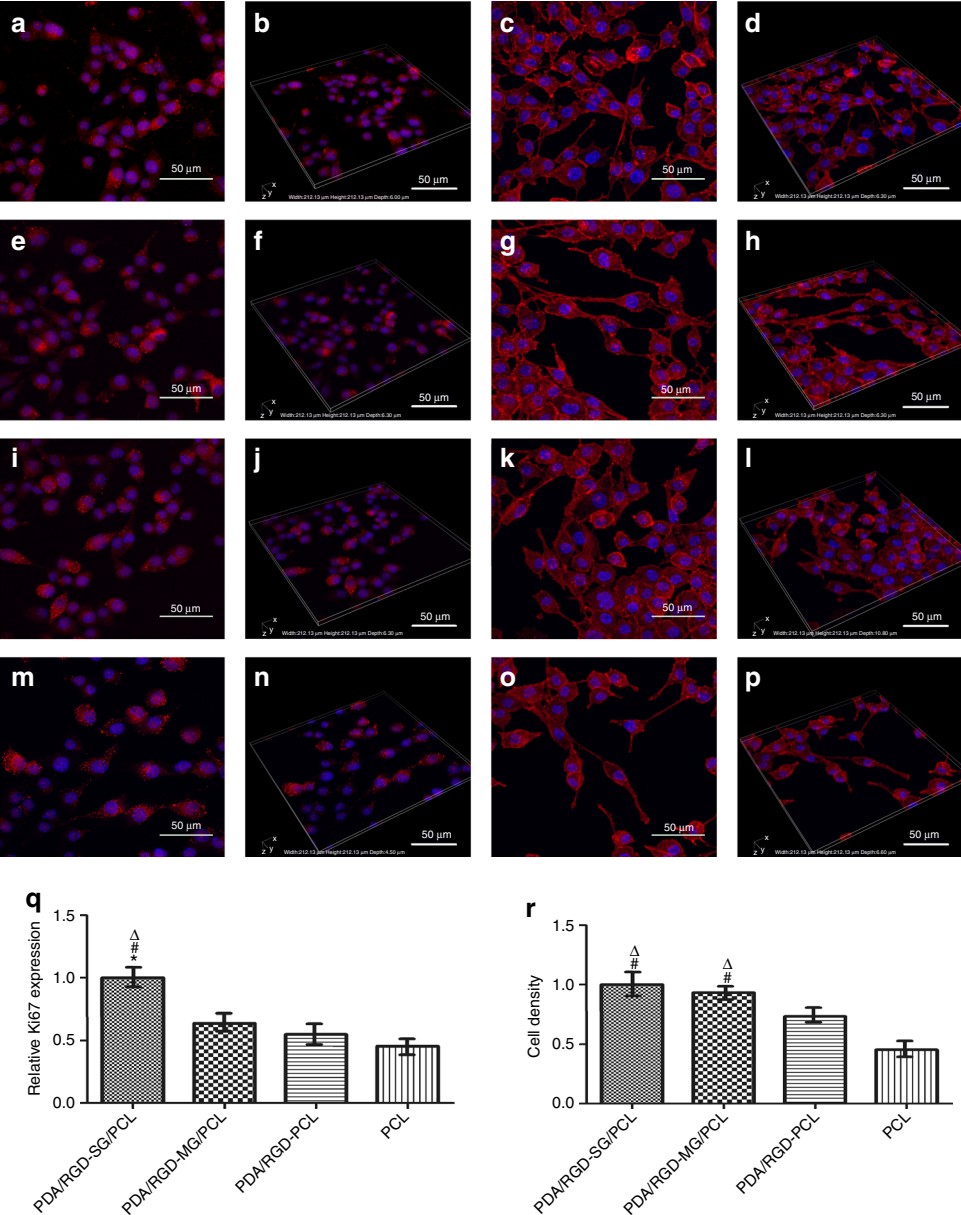

**Fig. 5** Immunofluorescent staining for Ki67 and F-actin. **a**, **b** Ki67 expression of SC on PDA/RGD-SG/PCL. **e**, **f** Ki67 expression of SC on PDA/RGD-MG/PCL. **i**, **j** Ki67 expression of SC on PDA/RGD-PCL. **m**, **n** Ki67 expression of SC on PCL. **c**, **d** Phalloidin staining on PDA/RGD-SG/PCL. **g**, **h** Phalloidin staining on PDA/RGD-MG/PCL. **k**, **l** Phalloidin staining on PDA/RGD-PCL. **o**, **p** Phalloidin staining on PCL. **q** Relative expression of Ki67. **r** Cell density evaluation from phalloidin staining. The scale bar is 50 μm. All data are displayed as mean ± standard deviation. *$p < 0.05$ compared with PDA/RGD-MG/PCL; #$p < 0.05$ compared with PDA/RGD-PCL; Δ$p < 0.05$ compared with PCL (the statistical test is ANOVA)

**Nerve regeneration improvement of graphene conduits**. To evaluate morphological nerve regeneration and nerve expression, regenerated nerves were dissected immediately after electrophysiological analysis. Samples were processed by Hematoxylin & Eosin (HE) staining, 1% toluidine blue (TB) staining, and transmission electron microscopy (TEM). We displayed representative images of SC-loaded PDA/RGD-SG/PCL, SC-loaded PDA/RGD-MG/PCL, PDA/RGD-SG/PCL, PDA/RGD-MG/PCL, PDA/RGD-PCL, and autograft groups (Figs. 7 and 8, Supplementary Figs. 5–8). The appearance of regenerated axon fibers was exhibited and calculated via TB staining and TEM observation. Most of the regenerated nerves were well organized and lacked scar tissues. Four parameters were included for evaluation: number of myelinated axons, thickness of myelin sheath, regenerated axon area, and average myelinated axon diameter. The

number of myelinated axons was significantly higher in the autograft group, followed by Schwann cell-loaded PDA/RGD-SG/PCL, SC-loaded PDA/RGD-MG/PCL, PDA/RGD-SG/PCL, PDA/RGD-MG/PCL, and PDA/RGD-PCL groups. At 6 and 12 weeks, the results of SC-loaded PDA/RGD-SG/PCL and PDA/RGD-MG/PCL groups were significantly better than those of other groups (ANOVA, $p < 0.05$). But they were lower than that of the autograft group (ANOVA, $p < 0.05$). However, at 18 weeks, the value from SC-loaded PDA/RGD-SG/PCL and PDA/RGD-MG/PCL groups showed no significant differences from the autograft group (ANOVA, $p > 0.05$). So axon area was regenerated (Supplementary Fig. 9).

For axonal regrowth and nerve remyelination evaluation, Tuj1, NF200, S100, and myelin basic protein (MBP) were involved in immunofluorescence assay. We performed Tuj1 and NF200 triple

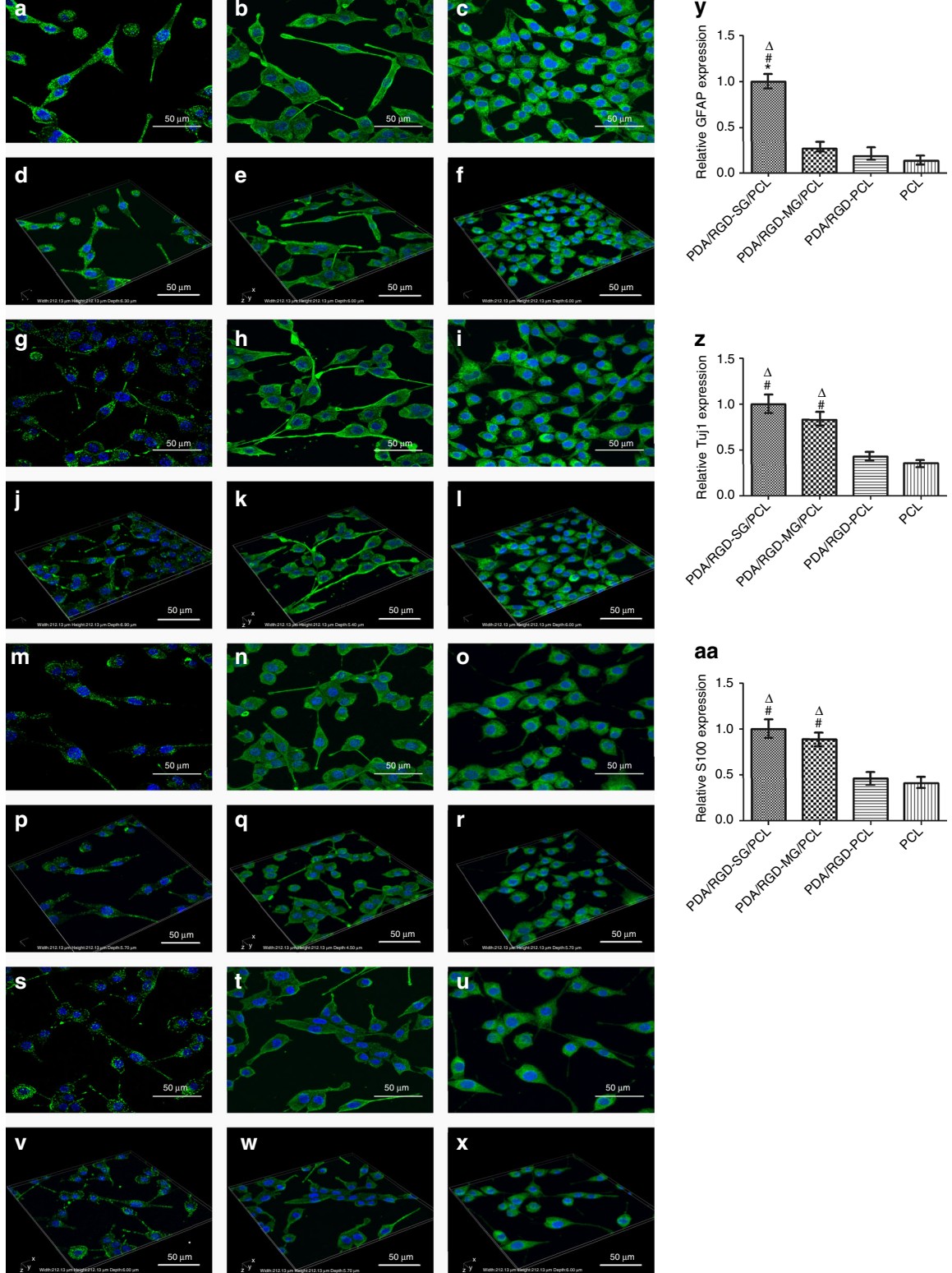

**Fig. 6** Immunofluorescent staining for GFAP, Tuj1, and S100. **a**, **d** GFAP expression of SC on PDA/RGD-SG/PCL. **g**, **j** GFAP expression of SC on PDA/RGD-MG/PCL. **m**, **p** GFAP expression of SC on PDA/RGD-PCL. **s**, **v** GFAP expression of SC on PCL. **b**, **e** Tuj1 expression of SC on PDA/RGD-SG/PCL. **h**, **k** Tuj1 expression of SC on PDA/RGD-MG/PCL. **n**, **q** Tuj1 expression of SC on PDA/RGD-PCL. **t**, **w** Tuj1 expression of SC on PCL. **c**, **f** S100 expression of SC on PDA/RGD-SG/PCL. **i**, **l** S100 expression of SC on PDA/RGD-MG/PCL. **o**, **r** S100 expression of SC on PDA/RGD-PCL. **u**, **x** S100 expression of SC on PCL. **y** Relative GFAP expression. **z** Relative Tuj1 expression. **aa** Relative S100 expression. The scale bar is 50 μm. All data are displayed as mean ± standard deviation. *$p < 0.05$ compared with PDA/RGD-MG/PCL; #$p < 0.05$ compared with PDA/RGD-PCL; Δ$p < 0.05$ compared with PCL (the statistical test is ANOVA)

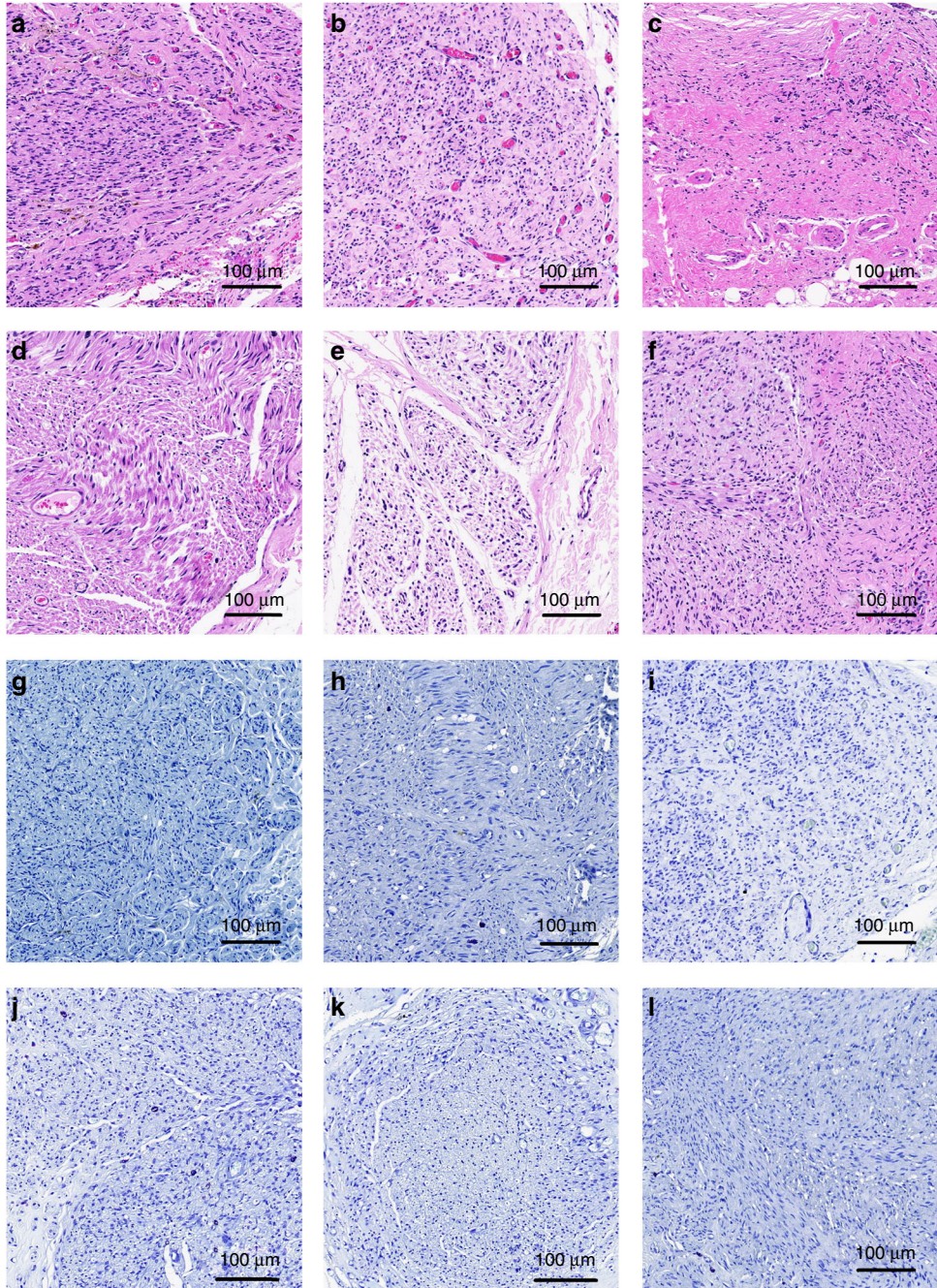

**Fig. 7** Nerve regeneration at 18 weeks postoperatively. HE (**a**–**f**) and TB (**g**–**l**) staining for regenerated nerves at 18 weeks post operatively. **a**, **g** SC-loaded PDA/RGD-SG/PCL. **b**, **h** SC-loaded PDA/RGD-MG/PCL. **c**, **i** PDA/RGD-SG/PCL. **d**, **j** PDA/RGD-MG/PCL. **e**, **k** PDA/RGD-PCL. **f**, **l** Autograft. The scale bar is 100 μm

staining as well as S100 and MBP triple staining. NF200 and Tuj1 represented regenerated neurofilaments and axons. S100 represented migration of Schwann cells and MBP indicated myelinated fibers. At 6 and 12 weeks after surgery, the autograft group showed better results in Tuj1, NF200, and S100 expression than all other groups (ANOVA, $p < 0.05$). At 18 weeks, the expression of Tuj1, NF200, and S100 was notably increased in SC-loaded PDA/RGD-SG/PCL and PDA/RGD-MG/PCL groups. It was significantly higher than that of PDA/RGD-SG/PCL, PDA/RGD-MG/PCL, and PDA/RGD-PCL groups (ANOVA, $p < 0.05$) and was slightly lower than that of the autograft group (ANOVA, $p > 0.05$). The MBP expression was generally low in each group at 6, 12, and 18 weeks (Figs. 9 and 10, Supplementary Figs. 10–13).

This indicated that PDA/RGD-SG/PCL and PDA/RGD-MG/PCL nerve conduits could promote nerve regeneration after long-term restoration in nerve conduit area and the effect was further enhanced by Schwann cell loading.

## Discussion

In this study, we fabricated PDA/RGD-coated macroporous single-layered and multi-layered graphene/PCL nerve conduits via the LBLC method. Unlike traditional electrospinning fabrication of nerve conduit, our 3D printing technique successfully avoided many problems, including failure in quality control, weak mechanical strength, random gaps between nanofibers, and most

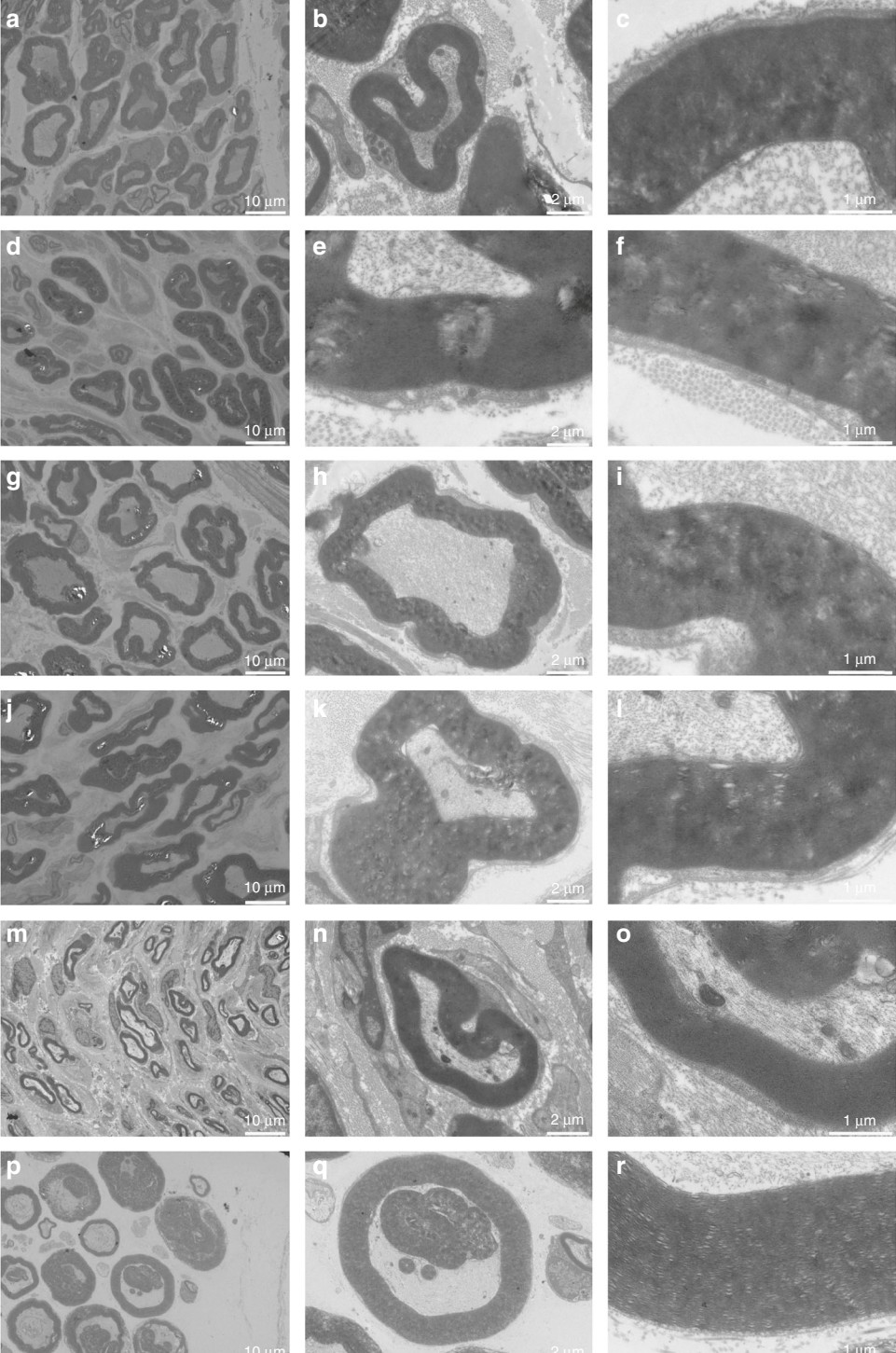

**Fig. 8** TEM for regenerated myelinated axons at 18 weeks post operatively. **a–c** SC-loaded PDA/RGD-SG/PCL. **d–f** SC-loaded PDA/RGD-MG/PCL. **g–i** PDA/RGD-SG/PCL. **j–l** PDA/RGD-MG/PCL. **m–o** PDA/RGD-PCL. **p–r** Autograft. The scale bar in **a**, **d**, **g**, **j**, **m**, and **p** is 10 μm. The scale bar in **b**, **e**, **h**, **k**, **n**, and **q** is 2 μm. The scale bar in **c**, **f**, **i**, **l**, **o**, and **r** is 1 μm

importantly uneven drug delivery distribution[22]. Via the LBLC method, 3D fabrication of nerve conduit allowed even distribution of graphene nanoparticles in the surface. This assured the ideal electric conductivity for peripheral nerve regrowth. Meanwhile, the 3D printer enabled us to add macroporous structures to the nerve conduit, which was vital for exchanges of nutrients and oxygen as well as endothelial cell proliferation. In addition, it is the first time that biodegradable materials have been used with

single-layered and multi-layered graphene for nerve conduit fabrication. PCL could successfully support the tubular structure for a long time. This was very significant for long-term peripheral nerve regeneration. In addition, PCL had appropriate stiffness and elasticity. Soft polymeric materials like collagen and chitosan can easily collapse and cause nerve contracture[28]. These properties guarantee a perfect scaffold material for ideal nerve function restoration.

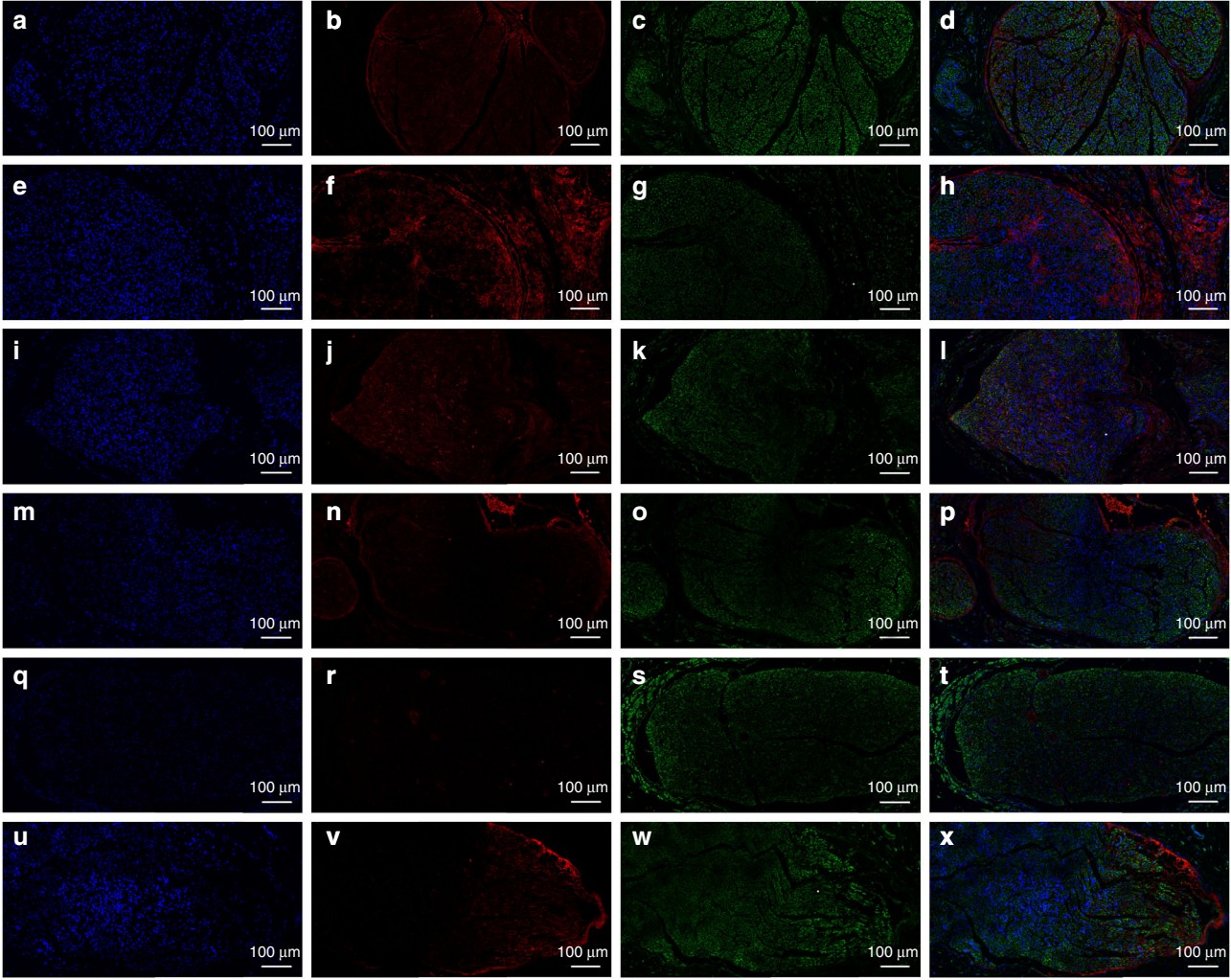

**Fig. 9** Triple immunofluorescent staining of Tuj1 and NF200 at 18 weeks post operatively. Tuj1 (green), NF200 (red), and nuclei (blue) were exhibited from different groups respectively. **a–d** SC-loaded PDA/RGD-SG/PCL. **e–h** SC-loaded PDA/RGD-MG/PCL. **i–l** PDA/RGD-SG/PCL. **m–p** PDA/RGD-MG/PCL. **q–t** PDA/RGD-PCL. **u–x** Autograft. The scale bar is 100 μm

As a carbon allotrope, graphene has two dimensions, and honeycomb-like structure, consisting of π bond-associated $sp^2$ orbital hybridization[29]. Graphene and its derivatives have extraordinary thermal and electric conductivity and therefore they have wide electron distribution[30]. The electric conductivity is closely associated with the layer number of graphene[31]. Electrical characteristics become very complicated because of the addition of layers and improvement of graphene electronic construction[32]. Novoselov et al. reported that only single and few-layer graphene could be considered as thin structures and thus they had strong ability of electrical transports[33]. Geim et al. mentioned that only single- and double-layered graphene could be regarded as semi-metals because they had no overlaps. Multi-layered structure could compromise electric conductivity. When it reached ten layers, the physical and chemical characteristics would be close to those of graphite[34]. In this study, the difference of carbon structure was also evaluated by Raman spectrum. Higher 2D peak was observed in single-layered graphene. While, lower 2D peak in comparison with G peak was observed in multi-layered graphene.

Actually, electrically conductive materials usually rely on external stimulation from the environment to improve nerve cell growth, attachment, and differentiation. Heo et al. fabricated graphene/polyethylene terephthalate film for nerve differentiation under electrical stimulation. In a noncontact method, they successfully strengthened cell–cell communication. For the underlying mechanism, the intercellular coupling was changed with alterations of endogenous cytoskeletal proteins[35].

Apart from electrical stimulation, there are many other methods for successful stimulation of neural stem cell differentiation and expression. Akhavan et al. discussed the use of nanoscale laser stimulation and graphene nanoscaffold in nerve differentiation for the first time. Pulsed laser could induce a relaxation with radial gradient and thermal stress. Therefore, it directed a certain differentiation of nerve stem cells. The self-organization style was very important in regeneration of the central nervous system[36]. Flash photo could induce electron accumulation on the surface of graphene membrane and initiated electric field. Akhavan et al. fabricated reduced graphene oxide and $TiO_2$ complex and repeatedly exerted flash stimulation to induce cells into neurons instead of glia cells. This highly accelerated proliferation and differentiation reaction mainly relied on Ti–C and Ti–O–C bonds[37]. Akhavan et al. believed that semiconductor was very important for successful stimulation. They focused on near infrared laser and discussed its stimulation effect on nerve stem cell differentiation. They observed better cell elongations and higher differentiation into neurons than reduced graphene oxide nanoscaffolds under moderate-energy photoelectron[38]. Tang et al. discovered chemical signals were also vital

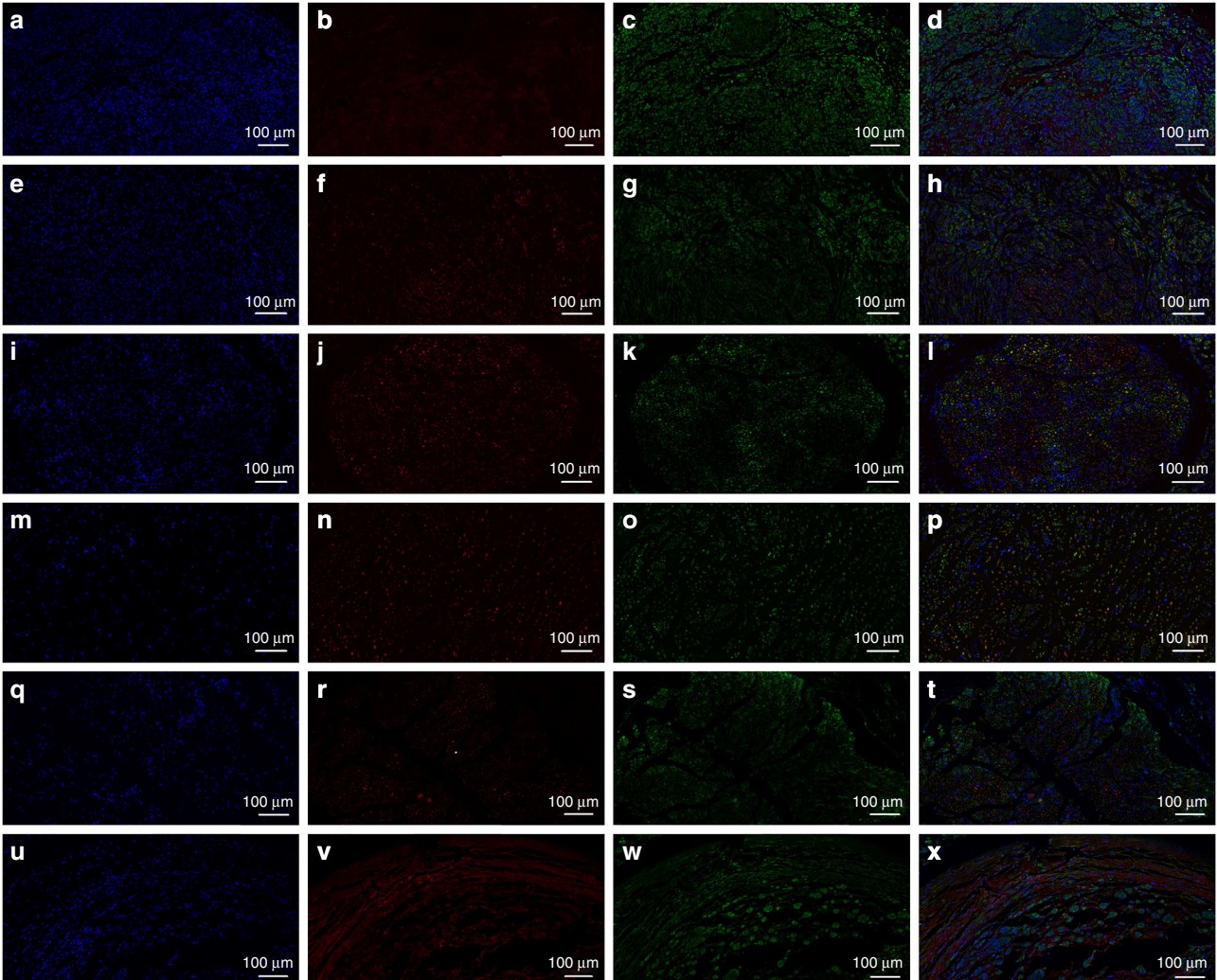

**Fig. 10** Triple immunofluorescent staining of S100 and MBP at 18 weeks post operatively. S100 (green), MBP (red), and nuclei (blue) were exhibited from different groups respectively. **a–d** SC-loaded PDA/RGD-SG/PCL. **e–h** SC-loaded PDA/RGD-MG/PCL. **i–l** PDA/RGD-SG/PCL. **m–p** PDA/RGD-MG/PCL. **q–t** PDA/RGD-PCL. **u–x** Autograft. The scale bar is 100 μm

to promoting electrical signaling of nerve stem cells with graphene nanoscaffold application. They applied high K$^+$ stimulation to cells and found it could upregulate intracellular calcium expression and stimulate massive activation of C-jun for neural network regulation[39].

Apart from the above stimulation, morphological modification was also vital to stem cell fate. Cells could sense biophysical cues in the microenvironment and experience cell–substrate reaction. Wang et al. fabricated a fluorinated graphene nanoscaffold and created microchannels between polydimethylsiloxane. They noticed an increase in neural expression compared with non-patterned nanoscaffolds[40].

Our 3D PDA and RGD coated single-layered and multi-layered graphene poly-caprolactone scaffolds were able to provide ideal cell signals including chemical signals from PDA and RGD coating and electrical signals from graphene conductivity of bioelectricity. Previously, Egeland et al. reported that they used poly(3,4-ethylenedioxythiophene)-modified neural interfaces to conduct action potential and stimulate biophysiological reaction in a severe nerve defect[41]. In addition, the free electrons from the environment generated some current through graphene material[42]. The macroporous and nanoporous structures equipped 3D graphene-based materials with special topological properties for

cell growth and attachment because they offered free entrance of water, protein, and other nutrients. An appropriate combination of morphological and bioelectrically conductive stimuli was used in the fabrication of our graphene-based materials. We focused on the performances of single-layered and multi-layered graphene/PCL conduits in sciatic nerve repair. They could both conduct bioelectrical signals and contribute to nutrition exchange and waste excretion by multi-porous structures.

The graphene is considered as the best carbon-based material with good biocompatibility in the nerve tissue engineering[43]. However, it still has minor cytotoxic effects on nerve cells. Therefore, we used adhesive macromolecules PDA and RGD to further improve the biocompatibility of graphene materials in this study. PDA can protect materials from potential threats and encapsulate small molecules in drug delivery[44]. In addition, it also works as a substrate or biomacromolecule[45]. In the past, biomaterial surface modification is usually referred to ECM-like proteins, including fibronectin[46], collagen[47], and laminin[48]. However, overlook in strict purification could lead to infection and immune responses[49]. Moreover, proteins were likely to undergo degradation in changeable in vivo environment. Therefore, protein coating was good but it was far from ideal. Peptides, in contrast, are stable under different sterilization, pH, and

temperature[50]. In addition, they are easily acquired and characterized. This makes them cost effective. Among various adhesive-associated peptides, RGD is commonly applied in tissue regeneration, coating chitosan[51], fibroin[52], titanium[53], and so forth. In the field of peripheral nerve injury, some literature has mentioned its advantageous role in nerve repair. Zhu et al. reported that RGD and YIGSR (a laminin oriented peptide which can promote neurite outgrowth) could be mixed together and used for lengthy sciatic nerve defects[54]. Yan et al. discovered satisfactory nerve regrowth by RGD and NGF modification for Poly (L-lactic acid-co-Ɛ-caprolactone) conduit[55]. However, the neural protective and repairing effect of these nerve conduits could not match the performance of autologous nerve transplant.

This is the first study to combine PDA and RGD in the modification of graphene/PCL nanoscaffolds in the field of peripheral nerve regeneration. The intended effect was to preserve the moderate electric conductivity of graphene and to significantly improve the biocompatibility of nanoscaffolds both in vitro and in vivo.

In vitro studies showed that PDA/RGD-SG/PCL and PDA/RGD-MG/PCL nanoscaffolds were capable of promoting SC attachment and proliferation. The proliferative ability did not show any significant differences in LIVE/DEAD cell staining. At the same time, in CCK8 assay, cells on PDA/RGD-SG/PCL and PDA/RGD-MG/PCL nanoscaffolds proliferated similarly like PDA/RGD-PCL and TCP, which revalidated the positive role of PDA/RGD in cell proliferation. Cells were attached firmly to different scaffolds via SEM observation. In addition, the porous structure of graphene allowed uniform distribution of nutrients like water, proteins, and growth factors to be absorbed freely by RSCs. The adhesive and proliferative ability of graphene was further evaluated by different protein expression. Both PCR and WB assays indicated SCs seeded on PDA/RGD-SG/PCL and PDA/RGD-MG/PCL nanoscaffolds secreted more adhesion-associated and proliferation-related proteins after co-culture for 3 days. The potential reason why graphene was beneficial to SC biological function was that moderate π–π bonds in graphene could help to improve metabolic activity and benefit cell growth and adhesion[56].

At the same time, PDA/RGD-SG/PCL and PDA/RGD-MG/ PCL nanoscaffolds both improved SC neural expression by increased expression of multiple neural-specific proteins, including GFAP and Tuj1. Apart from these two proteins, NGF, BDNF, GDNF, and CNTF have profound implications in peripheral nerve regeneration. The expression of NGF, BDNF, GDNF, and CNTF on PDA/RGD-SG/PCL nanoscaffolds was apparently higher than that on other nanoscaffolds. According to the results above, we were very positive about the fact that graphene could promote cell neural expression and single-layered graphene had more outstanding performance in bioelectrical signal transduction with its unique dimensional properties and stiff surface characteristics[57].

This is also the first study to evaluate long-term in vivo performance of graphene-based nerve conduit in peripheral nerve restoration. Although ideal biocompatibility was previously reported in graphene-related nerve scaffold in vivo by Jakus et al., they only inserted the nerve conduits subcutaneously for 1 month without long-term evaluation of nerve functional, morphological, and neural expression performance[58]. Our in vivo studies exhibited that both PDA/RGD-SG/PCL and PDA/RGD-MG/PCL nerve conduits contributed to functional sciatic nerve recovery and axon regrowth. Its effect was further enhanced by Schwann cell loading. Graphene could enhance cell migration and prolong SC viability seeded on the conduit for extended effects on nerve repair. In the meantime, graphene improved axon regrowth by increasing its quantity, thickness, and area with its porous

construction. The incorporation of graphene and PCL was also beneficial to decreasing PCL electrical resistance and enhancing neurite outgrowth. As for the biodegradation of graphene-based PCL conduit in vivo, we did not observe complete degradation at 18 weeks post operatively but only noticed the conduit was much softer than it was at implantation. It was previously reported that graphene in blood circulation could effectively be cleared from the body through renal excretion[59, 60]. In addition, the substrate material PCL generally degraded fully within 6–12 months in vivo[61]. Thus, with the slow degradation of PCL substrate, single-layered or multilayered graphene could barely cause toxic effects, especially at a relatively low concentration like 1% graphene in the PCL.

All the results indicated a positive role of PDA/RGD-SG/PCL and PDA/RGD-MG/PCL nanoscaffolds in the process of peripheral nerve regeneration, which was displayed by the promotion of axonal regeneration and functional nerve recovery. The graphene-based nanomaterials will be beneficial to curing long-range nerve defects in the future.

Via an integrated 3D printing fabrication and LBLC method, the nanoscaffolds exhibited excellent SC adhesion and proliferation performance. At the same time, they improved SC neural expression significantly in vitro. For in vivo evaluation, the PDA/RGD-SG/PCL and PDA/RGD-MG/PCL nerve conduits were also beneficial to promoting functional nerve recovery and axon regrowth in a 15-mm lengthy sciatic nerve defect. These results indicate that graphene-based nanotechnology will have great potential in peripheral nerve regeneration in preclinical and clinical application.

## Methods

**Graphene scaffold fabrication**. The single-layered graphene and multi-layered graphene were purchased from Suzhou Tanfeng Graphene Technology Co., Ltd. (China). According to the provider, the purity of single-layered graphene is over 98% and the purity of multi-layered graphene is over 95%. Detailed characteristics of single-layered graphene and multi-layered graphene are presented in Supplementary Figs. 14 and 15. PCL was purchased from Pertorp UK Limited Inc. PDA and RGD were purchased from Sigma-Aldrich. A 3D tabletop printer was used in the fabrication of graphene-based nanocomposite nerve conduit. Graphene nanoparticles and PCL were separately dissolved in dichloromethane in a tubular mold. The single-layered and multi-layered graphene/PCL solution was placed in a drying board and gradually formed a film shape. PDA and RGD were dissolved in water and vibrated for 10 min. The PDA/RGD mixture formed the conduit shape by doping with graphene/PCL film via the LBLC method to. Finally, after the conduit was solidified, the microneedles and the rolling tube mode were removed.

**Characterization of PDA/RGD-modified graphene nanoscaffolds**. The PDA/RGD-modified graphene nanoscaffold was examined using SEM for 3D structure evaluation. Samples were separated and prepared without conductive material coating. Cross sections from different samples were viewed with SEM (VEGA3 TESCAN) operating at an accelerating voltage of 5 kV. The pictures were taken at ×200, ×500, ×3000, and ×6000 magnifications. Random views were selected for final assessment. The single-layered and multi-layered carbon structure was evaluated using Raman spectroscopy (CRM 200 Witech, laser excitation at 532 nm). The surface elastic modulus was assessed using nanoindentation (Nano Indenter G200, Agilent, USA). The elastic recovery curves were recorded. At least six indents were recorded for final statistical evaluation. In addition, the conductive capability was measured using a four-point probe method by means of Hall Effect Measurement System (MMR, Yamingtec, China). We connected the probes to the two ends of the nerve conduits. PDA/RGD-modified single- and multi-layered graphene/PCL nanoscaffolds were prepared for conductivity assessment. Evaluation of both materials was repeated for five times in each experiment.

**Cell viability and attachment on graphene nanoscaffolds**. RSCs (ATCC CRL-2765) were purchased from the cell bank of the Chinese Academy of Sciences (Shanghai, China). We seeded RSCs on the different scaffolds and fixed these nanoscaffolds with a metal ring in the appropriate size. The metal ring and scaffolds were sterilized by exposure to ultraviolet light overnight. Different graphene/PCL was designed to determine the concentration dependent cytotoxicity on Schwann cells by CCK8 assay, including 0.1%, 0.5%, 1%, 2%, and 4% single- and multi-layered graphene in PCL. The experiment was repeated for five times.

SCs were cultured on PDA/RGD-SG/PCL, PDA/RGD-MG/PCL, PDA/RGD-modified PCL, and PCL nanoscaffolds for 24 h. Then, LIVE/DEAD cell kit

(Invitrogen, USA) was used for cell viability analysis according to the manufacturer's instructions. Calcein AM and Ethidium homodimer-1 were respectively used to measure intracellular esterase activity and plasma membrane integrity. They were added to medium-free cell/scaffolds after being mixed with Dulbecco's phosphate-buffered saline (DPBS). Before fluorescence microscope observation, we removed working solutions and washed the samples by DPBS for three times. The experiment was repeated for five times.

CCK8 was also used to evaluate cell proliferation. SCs were cultured in 90% high glucose DMEM with 10% fetal bovine serum (Gibco, USA) and 1% penicillin/streptomycin solution (Gibco, USA) for 1, 3, 5, and 7 days respectively at a starting density of $2\times10^4$ cm$^{-2}$. At each time point, CCK8 solution was added to cell/scaffold before co-culture for another 4 h. Then, relative proliferation state was evaluated using a microplate reader among PDA/RGD-SG/PCL, PDA/RGD-MG/PCL, PDA/RGD-PCL, and PCL measured by absorbance value. The experiment was repeated for five times.

We performed immunofluorescence and WB to further evaluate cell proliferation and attachment on the different scaffolds. Brdu and Ki67 were used to evaluate proliferative state. N-cadherin and vinculin were used to evaluate cell attachment. For immunofluorescence, cells were fixed by 4% paraformaldehyde for 30 min at room temperature after being washed with fresh PBS. Then we used 0.1% TritonX-100 and blocked the samples by 1% bovine serum albumin (BSA) under the same condition. Then we added the primary antibody Ki67 (1:250, Abcam, USA) and cultured it with samples at 4 °C overnight. On the second day, we added the anti-rabbit secondary antibody (1:2000, Abcam, USA) and cultured it with samples for 1 h at room temperature. We performed DAPI (1:500, Gibco, USA) staining with a drop of working solution. The samples were observed using a confocal immunofluorescence microscope (Leica, Germany). For WB, after cell lysis (Cell Signaling Technology, USA) and protein solubilization, the proteins were separated using gel electrophoresis. Proteins were then transferred from gels to polyvinylidene difluoride (Sigma Aldrich, USA). After blocking with non-fat dry milk (Sigma Aldrich, USA), samples were ready for incubation with primary antibodies including anti-Brdu (1:4000, Abcam, USA), anti-Ki67 (1:5000, Abcam, USA), anti-N-cadherin (1:3000, Abcam, USA), and anti-vinculin (1:5000, Abcam, USA) overnight with shake cultivation at room temperature. We added the anti-rabbit secondary antibodies (1:2000, Abcam, USA) and cultured it with samples for 1 h. Proteins were visualized using Amersham ECL Prime Western Blotting Detection Reagent (GE Healthcare). The experiment was repeated for five times.

**SC morphology on graphene nanoscaffolds**. SCs were seeded on the different nanoscaffolds for 3 days at a starting density of $4\times10^4$ cm$^{-2}$. Then, cell/scaffolds were carefully washed with PBS and fixed with 2.5% glutaraldehyde for 1 h. 1% osmium tetroxide was added as the secondary fixation for another 30 min. A graded ethanol was then performed for dehydration from 50 to 100% for 20 min. The samples were dried overnight after addition of hexamethyldisilazane (Sigma Aldrich, USA). Finally, a gold-layer coated sample was analyzed using SEM (VEGA3 TESCAN). The experiment was repeated for five times.

Besides SEM, phalloidin conjugated to Alexa Fluor 488 (1:200, Abcam, USA) was used for staining the actin filaments. The cells were fixed with 4% paraformaldehyde for 30 min at room temperature after being washed with fresh PBS. Then, 5 µl FITC-Phalloidin (Abcam, USA) and 150 µl PBS were mixed as working solution and stained cells at room temperature for 30 min. After that the samples were washed with PBS for three times. Then, we performed DAPI (1:500, Gibco, USA) staining with a drop of working solution. The samples were observed using a confocal immunofluorescence microscope (Leica, Germany). The experiment was repeated for five times.

**SC neural expression on graphene nanoscaffolds**. 95% SCs purity was evaluated by S100 staining, which was a specific marker of Schwann cells. To evaluate neural expression of SCs on the different nanoscaffolds, cell/scaffolds were co-cultured for 4 days at a starting density of $2\times10^4$ cm$^{-2}$ before immunofluorescence assay. Neural-specific markers including GFAP and Tuj1 were evaluated using immunofluorescent staining. Cells were fixed with 4% paraformaldehyde for 30 min at room temperature after being washed with fresh PBS. Then we used 0.1% TritonX-100 and blocked samples by 1% BSA under the same condition. Then we added the primary antibodies and cultured them with samples at 4 °C overnight. On the second day, we added the anti-rabbit secondary antibody (1:2000, Abcam, USA) and cultured it with samples for 1 h at room temperature. We performed DAPI (1:500, Gibco, USA) staining with a drop of working solution. We dropped the mounting solution (ProLong Antifade Reagents for Fixed Cells, ThermoFisher, USA) on the slides immediately to prevent quenching. We observed slides using a confocal immunofluorescence microscope (Leica, Germany). The experiment was repeated for five times. The primary antibodies were anti-S100 beta (1:100, Abcam, USA), anti-Tuj1 (1:500, Abcam, USA), and anti-GFAP (1:1000, Abcam, USA). The secondary antibody was Alexa Fluor 488-conjugated goat anti-rabbit IgG (1: 200, Gibco, USA). All immunofluorescent images were taken using confocal immunofluorescence microscope (Leica, USA). For further quantification analysis, we also performed WB and qPCR. Original blot image is shown in Supplementary Fig. 16. Tuj1 and GFAP were involved. Some nerve growth factors were included in qPCR. RNA was extracted from cell samples with TRizol agent (ThermoFisher, USA) according to the manufacturer's instructions. The cDNA was transformed

from mRNA with PrimeScriptTM Rtreagent Kit (Takara, Japan). And then real-time PCR was run on real-time PCR machine. The mRNAs for *BDNF, GDNF, CNTF, NGF*, and *β-actin* (internal control) were introduced below. The experiment was repeated for five times. Relative gene expression calculation was conducted by means of the $2^{-\Delta\Delta Ct}$. Sequences are *GDNF* front: GTAGGCCAGGCATGTTGC AG; *BDNF* front: AAAGAAGCAAACGTCCACGG; *CNTF* front: TTCTGCC TTTGCCTACCAGC; and *NGF* front: CACCCACCCAGTCTTCCACA.

**Animal surgery**. Ninety SD rats (male, weighing 150–200 g) were selected for in vivo experiment. They were randomly and evenly allocated into six groups, Schwann cell-loaded PDA/RGD-SG/PCL, Schwann cell-loaded PDA/RGD-MG/PCL, PDA/RGD-SG/PCL, PDA/RGD-MG/PCL, PDA/RGD-PCL, and autograft groups. Each group was evaluated at 6 weeks, 12 weeks, and 18 weeks post operatively. All animals were housed in a stable temperature and night/day cycle atmosphere with specific pathogen-free requirement. We injected 30 mg kg$^{-1}$ pentobarbital sodium into peritoneal cavity for anesthesia, exposure of right sciatic nerves was performed by the gluteal muscle incision, a 15-mm-long nerve was dissected and replaced with different 15-mm nerve conduits and autografts by 6–0 nylon sutures. Then, muscles and skins were successively sutured with 4–0 nylon sutures. For cell loading groups, SCs were pre-cultured on PDA/RGD-SG/PCL and PDA/RGD-MG/PCL nanoscaffolds at a density of $6\times10^6$ cm$^{-2}$ in the 5% CO$_2$ incubator overnight. All animals received penicillin injection intraperitoneally immediately after surgery for $10^5$ units. Observations and procedures were performed at 6, 12, and 18 weeks respectively. Animal care and use were in accordance with the guidelines of the Animal Ethics Committee for Shanghai Jiao Tong University. The biological replicates were spread equally among the groups compared and normal distribution was observed among the replicates. The Brown–Forsythe and Bartlett's tests showed no significant variation among groups.

**Functional analysis**. We performed walking track analysis to measure functional recovery. The following variables were included, the distance between the first toe and the fifth toe (TS), the third toe to the heal (PL), and the second toe to the fourth toe (IT). These indexes were measured and calculated based on experimental legs (E) and normal legs (N). We calculated SFI by the following formula. SFI = (−38.3 × (EPL−NPL)/NPL) + (109.5 × (ETS−NTS)/NTS) + (13.3 × (EIT −NIT)/NIT)−8.8. SFI was normally a negative number varying from 0 to −100. Zero means good function of the sciatic nerves. While −100 indicates complete loss of nerve function. The higher the number is, the better the function is. In addition, extensor postural thrust is an important indicator to evaluate the overall locomotor improvement. It is characteristic of postural reflex reaction and therefore it is associated with SFI. The body of the rat was protected by a surgical towel and the hind limbs was exposed. The injured leg was in contact with a digital metatarsus for 30 s, and the largest pushing force was recorded. The experiment was repeated for five times.

At 6, 12, and 18 weeks after surgery, the animals were sacrificed by over dosage anesthesia intraperitoneally. The gastrocnemius muscle of the experimental sides was removed for mass weighing using a digital scale (Sartorius, Germany). The experiment was repeated for five times. Apart from locomotor function recovery, we also evaluated sensory functional recovery from all the groups according to a previous research[62]. A paw withdrawal apparatus (Hargreaves Model 390, USA) was used to measure withdrawal latency induced by a specific radiant heat source. Rats were placed in the testing environment for 10 min before heating. Twenty five percent of the maximal heat for 0.1 s was exerted to rats with 10 min between each test. We only recorded data when a rat was stationary and standing on all four paws. The experiment was repeated for five times.

**Electrophysiological and histological analysis**. At 6, 12, and 18 weeks post operatively, SD rats were subjected to electrophysiological analysis. The right sciatic nerves were exposed again under anesthesia. Bipolar electrodes were fixed at the two ends of regenerated nerves to deliver single electrical signals. We implanted an electrode at the belly of gastrocnemius muscle to record electromyography. We recorded the various latency and distance between two ends of stimulation to measure NCV and DCMAP. The experiment was repeated for five times.

Regenerated nerves were dissected immediately after electrophysiological evaluation. Samples were processed using HE staining, 1% TB staining, and TEM.

For histochemical staining, all nerve samples were fixed by 4% paraformaldehyde overnight, followed by 2% osmium tetroxide and cacodylate buffer process (Sigma Aldrich, USA). We performed HE and TB staining after samples were embedded. We counted and calculated the number and average diameter of myelinated fibers in the middle portion of the regenerated sciatic nerves under a light microscope (Leica, USA). Immunohistochemistry was performed on 4–5 sections from each sample and at least three samples from each group. Each experiment was performed five times on separate days. Axon area and myelin thickness were evaluated using TEM. 4% uranylacetate and lead staining working solution (Sigma Aldrich, USA) were used to prepare nerve samples, which were examined using TEM (Titan, China) with 80 kV voltage.

**Immunofluorescent assay of regenerated nerves**. At 6, 12, and 18 weeks post operatively, the regenerated nerves were harvested after SD rats were sacrificed.

Specimens of nerves were fixed by 4% paraformaldehyde at room temperature for 24 h. They were washed by PBS for three times. Then the nerve segments were fixed by 1% osmium tetroxide, dehydrated, and embedded in Epon812 (Electron Microscopy Sciences, USA) resin. The cross sections were cut at 4 mm thick (Leica EM UC 6 ultramicrotome) and mounted on gelatin pre-coated slides. All samples were evaluated by Tuj1/NF200 and S100/MBP triple immunofluorescent staining. The primary antibodies were anti-Tuj1 (1:100, Abcam, USA), anti-NF200 (1:250, Abcam, USA), anti-S100 beta (1:500, Abcam, USA), and anti-MBP (1:200, Abcam, USA). The slides were observed under an immunofluorescence microscope (Leica, USA). The experiment was repeated for five times.

**Statistical analysis**. All tests were repeated for five times and results were displayed as mean $\pm$ standard deviation. A $p$ value of 0.05 was considered significant by one-way ANOVA.

**Data availability**. The authors declare that all data supporting the findings of this study are available within the paper and its Supplementary Information files or are available from the authors upon request.

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

## Acknowledgements

The study was supported by the Projects of National Science Foundation of China (No.81373366), the Projects of National Science Foundation of Shanghai, China (15ZR1432500), Projects of the Shanghai Committee of Science and Technology, China (No. 12XD1403800) and Funds for Interdisciplinary Projects of Medicine and Engineering of Shanghai JiaoTong University (No. YG2015MS06, YG2017MS22, and YG2016QN22), National Science and Technology Major Projects for "Major New Drugs Innovation and Development" (2017ZX09101005-008-002), SUMHS seed foundation project (No. HMSF-16-21-010), Science and Technology Development Foundation of Pudong New District, Shanghai, China (PKJ2016-Y55). We appreciate the help from faculties of Instrumental Analysis Centre (IAC) of Shanghai Jiao Tong University.

## Author contributions

W.Y. conceived the initial idea and the conceptualization, participated in the data extraction and analysis, and revised the manuscript. Y.Q., X.Z., Q.H., W.C., H.L., and W.Y. conceived and participated in its design, searched databases, extracted and assessed studies and helped to draft the manuscript. Y.Q. wrote the manuscript. All authors read and approved the final manuscript.

## Additional information

**Competing interests:** The authors declare no competing financial interests.

