## [Peer Review File · Nature Communications]

Reviewers' comments:

Reviewer #1 (Remarks to the Author):

Functionalized graphene-based materials were used for regeneration of neuronal tissues, both in vitro and in vivo. The subject is highly interesting and suitable biological tests have been presented. However, the material characterization needs further improvement. There are also other points which should be addressed by the authors before further consideration. Therefore, I suggest major revision of the manuscript based on the following comments:

1. Figure 1 shows some layered morphologies with stacked layer with micro-scales. But no strong data was presented to confirm 3D graphene structure. Better SEM images with higher resolutions are required. The 3D graphene scaffolds should present the morphologies given in, e.g., [Sci. Rep., 2013, 3, 1604] for CVD-graphene and [CARBON 97 (2016) 71–77] for GO.

2. The Raman spectra of the samples should be given. In addition, the authors should give a complete discussion about the 2D peak. In fact, a discussion about single- and multi-layer properties of the graphene sheets must be given based on the shape, intensity and position of the 2D band. The following references can be useful for analysis of the 2D peak of the Raman spectra:

Phys Rev Lett 2006;97:187401,
Nano Lett 2007;7:2645–9,
Nano Lett 2008;8:36–41,
Nature 457 2009 706–710
CARBON 81 (2 0 1 5) 1 5 8 – 1 6 6

3. The graphene has been known as the best carbon-based degradable as well as biocompatible material for neuronal proliferation and differentiation. See, for example [J. Mater. Chem. B, 2016, 4, 3169–3190 and references therein] as a review in this regard.

4. It was stated that "Neural expression was significantly improved by electrically conductive 3D graphene scaffold", this can mean that electrical pulses were used for stimulation of the cells. But I could not find anything relating to, e.g., electrical stimulation of the cells. This should be clarified. In fact, stimulation is often required for neuronal growth. For example, electrical [Biomaterials, 2011, 32, 19–27], pulsed laser [J. Mater. Chem. B, 2014, 2, 5602–5611], flash photo [Nanoscale, 2013, 5, 10316–10326.], near infrared [Colloids Surf., B, 2015, 126, 313–321.], chemical [Biomaterials, 2013, 34, 6402–6411] and morphological [Adv. Mater., 2012, 24, 4285–4290.] stimuli. The authors should address to these methods and also clarify their method as compared to the previous known ones.

5. The manuscript is too length and a little boring (although the results are interesting).

Therefore, I suggest transferring some data and figures into Supplementary Information. The main results can be kept in the main stream of the text.

Reviewer #2 (Remarks to the Author):

The manuscript prepares multi-layered porous scaffold by 3D printing with layer-by-layer casting method. But the novelty of the manuscript is not enough. I think this manuscript is currently not meeting the standard of recommended journal. My major concerns are described below:

1. How the Schwann cells incubated with different graphene/PCL nanoscaffolds?
2. After promoting Schwann cells proliferation, were the designed graphene/PCL nanoscaffolds stay in the body?
3. How to confirm the graphene is single layered or four layered. To the best of my knowledge it is very difficult to find out since you used thousands of thousands pieces of graphene sheets, how to make sure all of them are single layered or four layered?
4. The authors claimed that the enhanced properties should be attributed to the good conductivity of graphene. The conductivity of the single-layered graphene/PCL 130 conduit is only 8.92×10^{-3} S/cm. I believe the metallic nanowires, like gold or silver nanorods etc may have much better conductivity than graphene. The authors should make comparison on it.
5. The authors claimed that the conductivity of single-layered graphene is much higher than the multi-layered counterparts. Actually graphite usually exhibited much higher electrical conductivity than most reported graphene materials. However, it displayed much better
6. In line 132-133 the authors claimed that the electrical conductivity than other conductive materials, this may not be true.
7. In line 287 the authors claimed that "This assured the balanced and strong electrical conductivity for peripheral nerve regrowth." I wonder if the authors like to say the balanced conductivity is better or the higher the better. It is not clear.
8. The printing skill is just a layer-by-layer printing, it is not necessary to use a 3D printer which is more complicated.
9. Regarding the description in line 347, the polarity of pi-pi bonds in graphene should be very weak.

Reviewer #3 (Remarks to the Author):

The authors report 3D-printing of a novel graphene based nerve conduit to improve peripheral nerve regeneration. While polydopamine (PDA) and arginylglycylaspartic acid (RGD) have been used previously for coating nerve guidance channels/scaffolds/conduits, both compounds have been combined in this study for the first time for coating of conduits to protect against cytotoxicity of single-layered and multi-layered graphene. Moreover, long-term in vivo studies to improve peripheral nerve recovery after injury using graphene

based conduits with or without Schwann cell seeding have not been performed previously. The authors report that the overall outcome of the Schwann cell loaded PDA/RGD-SG/PCL and PDA/RGD-MG/PCL resembles the outcome achieved with an autograft. This reported finding is remarkable since, to my opinion, no artificial scaffold was able to match the regenerative improvement seen with a peripheral nerve autograft with respect to axon elongation, myelination and functional outcome.

However, there are a number of questions and specific concerns to be raised.

Fig. 3E,3F Cytotoxicity assay: What are the units of measure at the Y-axis?

Fig. 10, F-6: the staining of (myelinated) axons is very weak, suggesting that the number of axons in the autograft is lower than in graphene conduits (F-1 and F-2). This is in contrast to the histogram in Fig. 9H and text line 261/262 that reads: "The number of myelinated axons was significantly higher in autograft group...".

Moreover, the staining used in this figure (HE, Toluidine Blue) to indicate myelinating axons is questionable. A specific antibody to stain (peripheral) myelinated fibers should be used, e.g. antibodies directed to P0 or MBP.

Fig. 11 (TEM): The TEM resolution is very poor. (a) The layers of the myelin sheath are not visible and (b) no basement membrane, a structural characteristic of Schwann cells, can not be seen.

Fig. 12: This figure is incomplete. Panels A6-R6 are missing.

While in line 218 "walking track analysis" is mentioned to evaluate functional recovery, no evidence besides measuring toe distance variables is provided for locomotor/walking improvement.

Line 271: "Tuj1 stood for migration of Schwann cells". Tuj1/ β -III tubulin is an axonal marker in peripheral nerve rather than a Schwann cell marker. Tuj1 staining of axons could explain why the two histograms in Fig. 12S and 12T look identical. One would expect this since both NF200 and Tuj1 are axonal markers. To specifically label Schwann cells an antibody directed to S100 β should be used.

Line 290: the statement "...first time that biodegradable materials have been used with single-layered and multi-layered graphene..." is not proven as no biodegradability has been shown in this study - at least not within one year.

Line 294/295: "These qualities make it the best scaffold material for ideal nerve function restoration." This statement regarding Schwann cell seeded graphene based conduits is not valid since no direct comparison with other scaffolds is shown.

Line 309/310: "Thus, extra electrical signals are needed to replace the damaged signaling transduction system for efficient nerve restoration". This statement indicates that electrical activity of the conduit could replace the interrupted signaling transduction in the injured nerve. This is very unlikely. To my opinion, there is no prove in the literature for this assumption.

Discussion line 372/373: "The graphene based nanomaterials bring hope and light to people for curing long-range nerve defexcts in the near future". This outlook is overstretching the data and misleading. As the axon bridging distance in the present paper was in the same range (1.5 cm) as in many other successful peripheral nerve regeneration studies using conduits. Whether or not graphene based conduits do better in long distance bridging remains to be seen. There is no evidence in the present paper for bridging longer gaps between nerve stumps than 1.5 cm.

An important question remains: Is there evidence for sensory recovery following grafting of graphene loaded nanoscaffolds?

Finally, there are many typing errors in the text.

Reviewers' comments:

Reviewer #1 (Remarks to the Author):

Functionalized graphene-based materials were used for regeneration of neuronal tissues, both in vitro and in vivo. The subject is highly interesting and suitable biological tests have been presented. However, the material characterization needs further improvement. There are also other points which should be addressed by the authors before further consideration. Therefore, I suggest major revision of the manuscript based on the following comments:

1. Figure 1 shows some layered morphologies with stacked layer with micro-scales. But no strong data was presented to confirm 3D graphene structure. Better SEM images with higher resolutions are required. The 3D graphene scaffolds should present the morphologies given in, e.g., [Sci. Rep., 2013, 3, 1604] for CVD-graphene and [CARBON 97 (2016) 71–77] for GO.

RE: Thank you for bringing us these important references. In our previous figures, we failed to provide clear SEM images to show the 3D structure of our graphene based materials. Therefore, we have taken the reviewer's opinion seriously and operated SEM experiments again to evaluate 3D graphene structure with better resolutions. Please refer to the revised manuscript and figures attached. From SEM pictures, we can see multi-layered 3D fabrication and nanoporous structure in the graphene based nerve conduit with different magnification. These two references helped us a lot when we confirmed the correct structure required in this study.

Thank you very much for your advices!

2. The Raman spectra of the samples should be given. In addition, the authors should give a complete discussion about the 2D peak. In fact, a discussion about single- and multi-layer properties of the graphene sheets must be given based on the shape, intensity and position of the 2D band. The following references can be useful for analysis of the 2D peak of the Raman spectra:

Phys Rev Lett 2006;97:187401,

Nano Lett 2007;7:2645–9,

Nano Lett 2008;8:36–41,

Nature 457 2009 706–710

CARBON 81 (2 0 1 5) 1 5 8 – 1 6 6

RE: Thank you so much for bringing us so many significant references in Raman spectra and 2D peak analysis. We have read them thoroughly and find that Raman spectra is very important for us to understand electronic structure of graphene based material. We have taken the reviewer's opinion seriously and added this experiment. The 2D peak and G peak are exhibited in our revised manuscript (Supplementary Fig. 1). The 2D peak is a single and sharp one in single-layered graphene sheet. While it is relatively low in multi-layered graphene sheet. The intensity shows that the two materials have similar intensity for 2D peak. The position of 2D band from single-layered graphene sheet is around 2676 cm^{-1} , while it is 2682 cm^{-1} for multi-layered graphene sheet. The Raman spectrum helped us distinguish the different carbon structure of single-layered and multi-layered graphene because the reduction in layers caused different electronic dispersions. Thank you very much for your advices!

3. The graphene has been known as the best carbon-based degradable as well as biocompatible material for neuronal proliferation and differentiation. See, for example [Mater. Sci. Eng., C, 2014, 45, 196 – 204. and references therein] as a review in this regard.

RE: Thank you so much. This is a very significant review in the field of graphene based tissue engineering. This review focused on potential bio-applications of graphene-based nanomaterials for the proliferation and differentiation of nerve stem cells with different stimulation, like electrical, laser, flash photo, infrared, chemical, and some morphological stimuli. The article further discussed biocompatibility and possible degradation in the application of neural cells. We feel sincerely sorry for not having included this wonderful review in our previous manuscript. In the revised version, we have added it and talked about this reference to prove the beneficial effects of our graphene materials. Thank you very much for your advices!

4. It was stated that "Neural expression was significantly improved by electrically conductive 3D graphene scaffold", this can mean that electrical pulses were used for stimulation of the cells. But I could not find any thing relating to, e.g., electrical stimulation

of the cells. This should be clarified. In fact, stimulation is often required for neuronal growth. For example, electrical [Biomaterials, 2011, 32, 19–27], pulsed laser [J. Mater. Chem. B, 2014, 2, 5602–5611], flash photo [Nanoscale, 2013, 5, 10316–10326.], near infrared [Colloids Surf., B, 2015, 126, 313–321.], chemical [Biomaterials, 2013, 34, 6402–6411] and morphological [Adv. Mater., 2012, 24,4285–4290.] stimuli. The authors should address to these methods and also clarify their method as compared to the previous known ones.

RE: Thank you very much for providing us with so many excellent references. These references cover a wide topic including electric, laser, flash photo, near infrared, chemical and morphological stimuli. They are important extra stimulation methods that have been successfully applied in tissue engineering. We have read and learnt these references carefully and find that electrically conductive materials usually rely on extra stimulation from the environment to improve nerve cell growth, attachment and differentiation.

Heo et al. fabricated graphene/polyethylene terephthalate film for nerve differentiation accompanied by electrical stimulation. In a noncontact method, they successfully strengthened cell-cell communication. The underlying mechanism might be the intercellular coupling was changed with alterations of endogenous cytoskeletal proteins [Biomaterials, 2011, 32, 19–27.].

Akhavan et al. discussed the use of nanoscale laser stimulation in nerve differentiation on graphene nanoscaffold for the first time. Pulsed laser could induce a relaxation with radial gradient and thermal stress. Thus, it led to a directional differentiation for nerve stem cells. The self-organization style was very important in regeneration of central nervous system [J. Mater. Chem. B, 2014, 2, 5602–5611.].

Flash photo could induce electron accumulation on the surface of graphene membrane and initiated electric field. Akhavan et al. fabricated reduced graphene oxide and TiO₂ complex and repeatedly exerted flash stimulation on cells to induce them into neurons instead of glia cells. This highly accelerated proliferation and differentiation reaction was mainly dependent on Ti–C and Ti–O–C bonds [Nanoscale, 2013, 5, 10316–10326.].

Akhavan et al. believed that semi-conductor was very important for successful stimulation. They focused on near infrared laser and discussed its stimulation effect on neural stem cell differentiation. It was noticed that better cell elongations and higher differentiation into neurons than the common reduced graphene oxide nanosheets under moderate-energy photoelectron [Colloids Surf., B, 2015, 126, 313–321.].

Tang et al. discovered chemical signals were also important and effective for promoting electrical signaling of neural stem cells along with graphene nanoscaffold application. They applied high K⁺ stimulation to cells and found it could upregulate intracellular calcium expression and stimulate massive activation of C-jun for neural network regulation [Biomaterials, 2013, 34, 6402–6411.].

Apart from the above stimulation, morphological modification was also vital for stem cell fate. Cells could sense biophysical cues in the microenvironment and

experience cell-substrate reaction. Wang et al. fabricated a fluorinated graphene nanoscaffold and created microchannels between polydimethylsiloxane. They found that neural expression was increased in comparison with non-patterned nanosheets [Adv. Mater., 2012, 24,4285–4290.].

Our three-dimensional PDA and RGD coated single-layered and multi-layered graphene poly-caprolactone scaffolds are able to provide ideal cell signals including chemical signals from RGD and PDA coating and electrical signals from graphene conductivity of bioelectricity. Previously, Egeland et al. reported that they used poly(3,4-ethylenedioxythiophene)-modified neural interfaces to conduct action potential and stimulate biophysiological reaction through a significant nerve defect. Besides, the free electrons from the environment creates certain current through graphene material [Adv. Mater., 2011, 23, H263–H267.]. In addition, the macroporous and nanoporous structures equip 3D graphene based materials with special topological characteristics for cell growth and attachment because they offer free entrance of water, protein and other nutrition molecules. Therefore, an appropriate combination of morphological and bio-electrically conductive stimuli was used in the fabrication of our graphene based materials. Thank you very much for your advices!

5. The manuscript is too length and a little boring (although the results are interesting). Therefore, I suggest transferring some data and figures into Supplementary Information. The main results can be kept in the main stream of the text.

RE: Thank you for your opinion! We have revised this manuscript and moved some parts to Supplementary Information according to your advices. The most important results still remain in the main body. Thank you very much for your advices!

Reviewer #2 (Remarks to the Author):

The manuscript prepares multi-layered porous scaffold by 3D printing with layer-by-layer casting method. But the novelty of the manuscript is not enough. I think this manuscript is currently not meeting the standard of recommended journal. My major concerns are described below:

1. How the Schwann cells incubated with different graphene/PCL nanoscaffolds?

RE: We cultivated Schwann cells in common tissue culture plate for at least 24 hours to have relatively good viability. After that, we enzymatically detached them from the dish using 1.25% trypsin/ethylenediaminetetraacetic acid (EDTA) solution and reseeded them on the graphene/PCL nanoscaffolds. The nanoscaffolds were designed with different sizes to fit 24-well plate, 6-well plate and 100-mm plate. Then, the nanosheets were sterilized with 4-hour immersion of ethyl alcohol and 4-hour ultraviolet light exposure. After that, we used PBS to wash the nanoscaffolds for a couple of times to remove ethyl alcohol. All the nanosheets were placed in the different plates and were fixed with a metal ring above, which fit

the size of different plates and were previously sterilized as well. Finally, the cells were seeded on the nanofibers at an appropriate density. Thank you very much for your advices!

2. After promoting Schwann cells proliferation, were the designed graphene/PCL nanoscaffolds stay in the body?

RE: Yes. As a matter of fact, the PCL substrate has a relatively low degradation rate and thus we use it for long-term *in vivo* experiments to offer a mechanical support for regenerated sciatic nerves. For *in vivo* experiments, graphene/PCL conduit has a stable degradation rate for approximately one year [J Macromol Sci Pure Appl Chem. 1994; 32: 867-873.]. We have performed 18-week *in vivo* experiments. From optical images, conduits have degraded to some extent. It was previously reported that graphene in blood circulation can be effectively cleared from the body through renal excretion [Nano Lett. 2010; 10: 3318–3323; Mater Sci Eng C 2014; 45: 196–204.]. Thus, it caused negligible toxic effects to the living body, especially at a low concentration like 1% graphene in the PCL. Thank you very much for your advices!

3. How to confirm the graphene is singled layered or four layered. To the best of my knowledge it is very difficult to find out since you used thousands of thousands pieces of graphene sheets, how to make sure all of them are single layered or four layered?

RE: Thank you for your valuable opinion! It is very true that we can hardly say our single-layered graphene or multi-layered graphene is 100% pure. Both single-layered and multi-layered graphene nanoparticles were purchased from a graphene company (Hengqiu Tech.Inc., China). According to the manufacturers certificate, the single-layered graphene is 97% pure while multi-layered graphene is 95% pure respectively. In order to distinguish single-layered from multi-layered graphene, we performed Raman spectrum and evaluated the morphology, intensity and position of 2D band [Phys Rev Lett 2006;97:187401, Nano Lett 2007;7:2645–9, Nano Lett 2008;8:36–41, Nature 2009; 457: 706–710, Carbon 2015; 81:158 –166.]. We intended to tell the difference and make thorough discussion accordingly. Thank you very much for your advices!

4. The authors claimed that the enhanced properties should be attributed to the good conductivity of graphene. The conductivity of the single-layered graphene/PCL130 conduit is only 8.92×10^{-3} S/cm. I believe the metallic nanowires, like gold or silver nanorods etc may have much better conductivity than graphene. The authors should make comparison on it.

RE: We are sorry for making this assumption based on present lab results. It is very true that our single-layered and multi-layered graphene/PCL nanoscaffolds have a much weaker electric conductivity than previously reported graphene nanomaterials or the metallic nanowires, like gold or silver nanorods etc. This is due to the low concentration of graphene nanoparticles in the conduit. Therefore, many metallic nanowires can have better conductivity than our graphene based

materials.

However, if we compare pure graphene nanoparticles or nanosheets with pure gold nanorods or silver nanowires, we can find out that graphene shows wonderful electron mobility with more than $15000 \text{ cm}^2 \cdot \text{V}^{-1} \cdot \text{s}^{-1}$ and it has higher electric conductivity, better than copper or silver [JETP Lett. 1985; 42: 257–260; Nature 2005; 438: 197; Nat. Mater. 2007; 6: 183; Nat. Nanotech. 2008; 3: 206–9; J. Appl. Phys. 2008; 103, 053702; Nat. Nanotech. 2010; 5: 487–496; Rep. Prog. Phys. 2011; 74: 082501; Mater. Sci. Eng., C, 2014; 45: 196–204.]. In addition, silver nanowire composite displayed better electric conductivity than gold nanorod based scaffold. Therefore, graphene should have a better performance in electric conductivity than silver, copper and gold [ACS Nano, 2013; 7: 851–856; Acta Biomater. 2016; 41:133-46.]. Moreover, pure gold nanorods or silver nanowires cannot be fabricated into a nerve conduit because they cannot physically link the nerve ends. More importantly, they have very high toxicity [Small. 2012; 8: 1270–1278; Food and Chemical Toxicology 2014; 67: 80–86.]. They have to be mixed with biodegradable scaffolds, such as PCL, PLCL, and PLLA.

In order to compare our graphene based PCL nerve scaffolds with other metallic nanowires, such as gold nanorods or silver nanowires, we made comparison on it with some experiments. We fabricated PDA/RGD modified gold nanorods or silver nanowires based PCL scaffolds of different concentrations, including 0.5%, 1%, 2%, and 4%. The cytotoxicity of different scaffolds was displayed as follows. The results from 1% gold nanorods or silver nanowires based PCL scaffolds were slightly better than 0.5% gold nanorods or silver nanowires based PCL scaffolds, and they were significantly better than the rest. Cell viability was extremely low in 4% gold nanorods or silver nanowires based PCL scaffolds, indicating their potential cytotoxicity to Schwann cells (Peer review file figure 1). This confirmed that gold nanorods or silver nanowires based PCL scaffolds at low concentration could have minimal cytotoxicity. gold nanorods or silver nanowires at higher concentration would be very harmful to cell viability although they exhibited much better electric conductivity.

Peer Review File Figure 1. Cell viability assay of LIVE/DEAD cell staining and CCK8. (a,b) Live/dead pictures for PDA/RGD-gold/PCL. **(c,d)** Live/dead/merge pictures for PDA/RGD-silver/PCL. **(e,f)** Live/dead/merge pictures for PDA/RGD-graphite/PCL. The scale bar is 50 μ m. **(g)** Cytotoxicity assay for 0.5%, 1%, 2%, and 4% gold/PCL at different time points. **(h)** Cytotoxicity assay for 0.5%, 1%, 2%, and 4% silver/PCL at different time points. **(i)** Cytotoxicity assay for 0.5%, 1%, 2%, and 4% graphite/PCL at different time points. All data are displayed as mean \pm standard deviation. [#]p<0.05 compared with 2% gold(silver/graphite)/PCL; [^]p<0.05 compared with 4% gold(silver/graphite)/PCL. **(j)** Relative cell viability by LIVE/DEAD cell staining. **(k)** CCK8 assay for all groups.*p<0.05 compared with PDA/RGD-gold/PCL; [#]p<0.05 compared with PDA/RGD-silver/PCL; [^]p<0.05 compared with PDA/RGD-graphite/PCL; ^op<0.05 compared with PDA/RGD-PCL; ^qp<0.05 compared with PCL; ^ψp<0.05 compared with TCP (the statistical test is ANOVA).

We also performed characterization of PDA-/RGD-gold/PCL, and PDA/RGD-silver/PCL scaffolds. We evaluated the surface structure of gold nanorods, silver nanowires as well as gold nanorods or silver nanowires based PCL scaffold morphology using SEM. The results confirmed the nanomaterials we used were gold nanorods and silver nanowires. In addition, the scaffold displayed similar multi-layer and porous structure like PDA/RGD-SG/PCL and

PDA/RGD-MG/PCL scaffolds. We also evaluated the scaffold thickness, elastic modulus and electric conductivity of these materials (Peer review file figure 2). The electric conductivity of PDA/RGD-gold/PCL and PDA/RGD-silver/PCL scaffolds was much worse than PDA/RGD-SG/PCL and PDA/RGD-MG/PCL scaffolds at the same concentration of nanomaterials in the scaffold. This indicated that SG and MG had better electric conductivity than gold nanorod and silver nanowires in the PCL scaffold.

In addition, we performed CCK8 assay to evaluate the proliferative effects of different scaffolds on RSCs. After 168h, cells showed the best proliferation state on PDA/RGD-SG/PCL and PDA/RGD-MG/PCL scaffolds, which were far better than gold nanorods and silver nanowires based PCL scaffolds. We reached similar conclusions from LIVE/DEAD cell staining, and Ki67 immunofluorescent staining (Peer review file figures 1 and 2). In addition, phalloidin staining showed that cell density was much lower on gold nanorods or silver nanowires based PCL scaffolds in comparison with SG and MG based scaffolds. The protuberances were more extended on graphene based PCL scaffolds than gold and silver based scaffolds (Peer review file figure 3 and 4). This further indicated the gold nanorods or silver nanowires based PCL scaffolds were not as good as graphene/PCL scaffolds in supporting cell attachment and proliferation.

g	Scaffold thickness (mm)	Elastic modulus (MPa)	Electric conductivity ($S\ cm^{-1}$)
PDA/RGD-gold/PCL	0.45	54.25	5.89×10^{-4}
PDA/RGD-silver/PCL	0.46	51.68	8.76×10^{-4}
PDA/RGD-graphite/PCL	0.48	57.97	1.45×10^{-4}

Peer Review File Figure 2. Characterization of gold nanorods, silver nanowires, PDA/RGD-gold/PCL, PDA/RGD-silver/PCL and PDA/RGD-graphite/PCL nerve conduits. SEM images for structural characteristics of gold nanorods and silver nanowires (a,b) and evaluation of the nanoporous and multi-layered 3D structure in PDA/RGD-gold/PCL, PDA/RGD-silver/PCL and PDA/RGD-graphite/PCL nerve conduits (c-f). Thickness, elastic modulus, and electric conductivity of PDA/RGD-gold/PCL, PDA/RGD-silver/PCL and PDA/RGD-graphite/PCL nerve conduits (g) (Evaluation of all materials was repeated for five times).

Peer Review File Figure 3. Immunofluorescent staining for Ki67. (a-c) Ki67 expression of SC on PDA/RGD-gold/PCL. (d-f) Ki67 expression of SC on PDA/RGD-silver/PCL. (g-i) Ki67 expression of SC on PDA/RGD-graphite/PCL. (j) Relative expression of Ki67. The scale bar is 50 μm . All data are displayed as mean \pm standard deviation. * $p < 0.05$ compared with PDA/RGD-MG/PCL; # $p < 0.05$ compared with PDA/RGD-gold/PCL; $\Delta p < 0.05$ compared with PDA/RGD-silver/PCL; $\circ p < 0.05$ compared with PDA/RGD-graphite/PCL; $\phi p < 0.05$ compared with PDA/RGD-PCL; $\psi p < 0.05$ compared with PCL (the statistical test is ANOVA).

Peer Review File Figure 4. Phalloidin staining of RSC on different nanoscaffolds. (a-c) Phalloidin staining on PDA/RGD-gold/PCL. **(d-f)** Phalloidin staining on PDA/RGD-silver/PCL. **(g-i)** Phalloidin staining on PDA/RGD-graphite/PCL. **(j)** Cell density evaluation from phalloidin staining. The scale bar is 50 μm . All data are displayed as mean \pm standard deviation. # $p < 0.05$ compared with PDA/RGD-gold/PCL; $\Delta p < 0.05$ compared with PDA/RGD-silver/PCL; $\theta p < 0.05$ compared with PDA/RGD-graphite/PCL; $\phi p < 0.05$ compared with PDA/RGD-PCL; $\varphi p < 0.05$ compared with PCL (the statistical test is ANOVA).

These results validated the statement that under the same concentration, 1% SG and MG in PCL scaffolds can better improve cell viability than 1% gold nanorods or 1% silver nanowires based PCL scaffolds, due to their electric conductivity, and micropatterning.

Apart from the cell viability, we also evaluated the potential effects of gold nanorods or silver nanowires based PCL scaffolds on RSCs proliferation, attachment and neural expression. The immunofluorescent staining of S100, Tuj1 and GFAP was performed respectively. The results showed that 1% SG and MG based PCL scaffolds could better improve RSC neural expression than 1% gold

nanorods or silver nanowires based scaffolds (Peer review file figures 5-7). We further performed Western blotting assay and reached similar conclusions (Peer review file figure 8). We evaluated the Brdu, Ki67, N-cadherin, vinculin, GFAP and Tuj1 expression of RSCs cultured on different scaffolds. The results from PDA/RGD-SG/PCL and PDA/RGD-MG/PCL scaffolds were better than the others. From *in vitro* results, we could reach to this conclusion that 1% SG and MG based PCL scaffold showed better performance than 1% gold nanorods or silver nanowires based PCL scaffold in electric conductivity and promoting cell proliferation, attachment, and neural expression. We previously performed *in vivo* peripheral nerve regeneration experiments with 1% PDA/RGD-gold/PCL nerve conduit. At 18 weeks after injury, we noticed that sciatic nerves from the gold based PCL conduit displayed worse myelin sheath thickness than our SG and MG based PCL conduit via TEM observation in this study (The results are not published. Peer review file figure 9).

Peer Review File Figure 5. Immunofluorescent staining for S100. (a-c) S100 expression of SC on PDA/RGD-gold/PCL. (d-f) S100 expression of SC on PDA/RGD-silver/PCL. (g-i) S100 expression of SC on PDA/RGD-graphite/PCL. (j) Relative expression of S100. The scale bar is 50 μm. All data are displayed as mean ± standard deviation. #p<0.05 compared with PDA/RGD-gold/PCL; ^p<0.05 compared with PDA/RGD-silver/PCL; °p<0.05 compared with PDA/RGD-graphite/PCL; φp<0.05 compared with PDA/RGD-PCL; φp<0.05 compared with PCL (the statistical test is ANOVA).

Peer Review File Figure 6. Immunofluorescent staining for GFAP. (a-c) GFAP expression of SC on PDA/RGD-gold/PCL. (d-f) GFAP expression of SC on PDA/RGD-silver/PCL. (g-i) GFAP expression of SC on PDA/RGD-graphite/PCL. (j) Relative expression of GFAP. The scale bar is 50 μm. All data are displayed as mean ± standard deviation. * $p < 0.05$ compared with PDA/RGD-MG/PCL; # $p < 0.05$ compared with PDA/RGD-gold/PCL; Δ $p < 0.05$ compared with PDA/RGD-silver/PCL; Φ $p < 0.05$ compared with PDA/RGD-graphite/PCL; † $p < 0.05$ compared with PDA/RGD-PCL; Ψ $p < 0.05$ compared with PCL (the statistical test is ANOVA).

Peer Review File Figure 7. Immunofluorescent staining for Tuj1. (a-c) Tuj1 expression of SC on PDA/RGD-gold/PCL. (d-f) Tuj1 expression of SC on PDA/RGD-silver/PCL. (g-i) Tuj1 expression of SC on PDA/RGD-graphite/PCL. (j) Relative expression of Tuj1. The scale bar is 50 μm. All data are displayed as mean ± standard deviation. #p<0.05 compared with PDA/RGD-gold/PCL; ^p<0.05 compared with PDA/RGD-silver/PCL; °p<0.05 compared with PDA/RGD-graphite/PCL; °p<0.05 compared with PDA/RGD-PCL; ϕp<0.05 compared with PCL (the statistical test is ANOVA).

Peer Review File Figure 8. WB assay of Ki67, Brdu, GFAP, Tuj1, N-cadherin and vinculin. Their relative expression from SC seeded PDA/RGD-SG/PCL, PDA/RGD-MG/PCL, PDA/RGD-gold/PCL, PDA/RGD-silver/PCL, PDA/RGD-graphite/PCL, PDA/RGD-PCL and PCL nanoscaffolds (results were normalized to actin). From left to right in all the blots, they are TCP, PCL, PDA/RGD-PCL, PDA/RGD-silver/PCL, PDA/RGD-graphite/PCL, PDA/RGD-gold/PCL, PDA/RGD-MG/PCL, and PDA/RGD-SG/PCL groups. ϑ $p < 0.05$ compared with PDA/RGD-SG/PCL; θ $p < 0.05$ compared with PDA/RGD-MG/PCL; Ψ $p < 0.05$ compared with PDA/RGD-gold/PCL; ϕ $p < 0.05$ compared with PDA/RGD-graphite/PCL; β $p < 0.05$ compared with PDA/RGD-silver/PCL; Ω $p < 0.05$ compared with PDA/RGD-PCL; $\#$ $p < 0.05$ compared with PCL (the statistical test is ANOVA).

Peer Review File Figure 9. (a-d) TEM for regenerated myelinated axons from PDA/RGD-gold/PCL conduit group at 18 weeks postoperatively. The scale bar in a is 10 μm . The scale bar in b is 2 μm . The scale bar in c and d is 1 μm . (e) Thickness of myelin sheath. All data are displayed as mean \pm standard deviation. * $p < 0.05$ compared with autograft ($n = 5$); $\phi p < 0.05$ compared with PDA/RGD-gold/PCL ($n = 5$); # $p < 0.05$ compared with PDA/RGD-PCL ($n = 5$); $\Delta p < 0.05$ compared with PDA/RGD-MG/PCL ($n = 5$). $\phi p < 0.05$ compared with PDA/RGD-SG/PCL ($n = 5$) (the statistical test is ANOVA).

The electric conductivity is one of the vital factors for nerve tissue regeneration. And graphene of high concentration is likely to add significant cytotoxicity to tissues [Mater. Sci. Eng. C 2014; 45: 196–204.]. Therefore, we selected a relatively

conductive and biofriendly concentration of graphene for our experiment at the cost of certain electric conductivity. In our study, the graphene based material improved Schwann cell proliferation, attachment and neural expression due to its conductivity and excellent micropatterning. The previous discussion about excellent electric conductivity of graphene based materials was modified and replaced with discussion and comparison with different external stimulation in promoting nerve regeneration and differentiation. Thank you very much for your advices!

5. The authors claimed that the conductivity of single-layered graphene is much higher than the multi-layered counterparts. Actually graphite usually exhibited much higher electrical conductivity than most reported graphene materials. However, it displayed much better

RE: From some articles reported previously, single-layered graphene exhibited better electric conductivity than multi-layered graphene. Iqbal et al. discussed the electrical performance of single, bi and multilayered graphene. A decreasing tendency of electrical transport was noticed with the addition of graphene layers [Sci Technol Adv Mater. 2014;15: 055004.]. Goenka et al also reviewed that single-layered graphene had better electric conductivity. With the addition of layers especially more than 10 layers, the characteristics of graphene will resemble graphite, which shows less satisfactory conductivity than graphene [J Control Release 2014; 173:75–88; Progress in Polymer Science 2010; 35:1350–1375; Progress in Materials Science 2011; 56: 1178–1271. CARBON 2010; 48: 2825-2830.].

Table 2
Material and compact characteristics.

Filler	BET surface area (m ² /g)	Conductivity (S/m)			
		Powder compact at 5 MPa	Paper	Isolated Single particle conductivity	Filler contribution limit from compact ^a
MWCNTs	272	5.43×10 ²	5×10 ³	10 ⁶ –10 ^{7b}	10.3×10 ³
Graphene	180	2.62×10 ²	1.4×10 ³	10 ⁷ –10 ⁸ [2]	10.9×10 ³
Carbon Black	56.9	5.58×10 ²	9×10 ¹	10 ³ [14]	8.8×10 ³
Graphite	3.08	2.12×10 ³	1.2×10 ³	10 ⁵ [27] ^c	13.8×10 ³

^a Estimate based on the model as described in [17].

^b As provided by the Nanocyl Company: <http://www.nanocyl.com/en/CNT-Expertise-Centre/Carbon-Nanotubes>.

^c Value highly variable depending on source.

In this table, the electric conductivity was compared among MWCNT (multiwall carbon nanotube), graphene and graphite. The conductivity of graphene was 10⁷–10⁸ S/m, much higher than the conductivity of graphite (10⁵ S/m) [Nature Nanotechnology 2008; 3: 206–209; Institute of Physics Handbook, 3rd ed. McGraw Hill, New York, 1972. Powder Technology 2012; 221: 351–358.].

Gao et al. fabricated a graphite nanoplatelet (GNP) polymer composite. The GNP content was 5.1 wt % in the polymer composite. And its electric conductivity was 3.8×10⁻³ S/cm [ACS Appl. Mater. Interfaces, 2013; 5:7758–7764.]. The electric

conductivity of PDA/RGD-SG/PCL and PDA/RGD-MG/PCL was 8.92×10^{-3} S/cm and 6.37×10^{-3} S/cm respectively in our study. And the graphene content in PCL scaffold was only 1%. This could also prove better electric conductivity of graphene than graphite.

In order to further discuss this problem, we fabricated 1% graphite based PCL conduit and compared its biocompatibility, cell proliferation, attachment, neural expression of RSCs *in vitro*.

Before that, we compared the cytotoxicity of 0.5%, 1%, 2% and 4% graphite/PCL scaffold on Schwann cells. The results showed that cell viability was the lowest in 4% graphite/PCL scaffold. 1% graphite/PCL scaffold was slightly better than 0.5% and much better than 2% graphite/PCL scaffold (Supplementary Fig. 1). This was consistent with the previous study that graphite had negative effects on cell proliferation due to its toxicity [Nanomedicine. 2015;10: 2423-2450.]. This confirmed that graphite based PCL scaffolds at low concentration could have minimal cytotoxicity. Graphite at higher concentration would be very harmful to cell viability although they exhibited much better electric conductivity.

In addition, we performed characterization of PDA-/RGD-graphite/PCL scaffolds and evaluated the surface morphology using SEM. The scaffold displayed similar multi-layer and porous structure like PDA/RGD-SG/PCL and PDA/RGD-MG/PCL scaffolds. We also evaluated the scaffold thickness, elastic modulus and electric conductivity of these materials (Peer review file figure 2). The elastic modulus and scaffold thickness of PDA/RGD-graphite/PCL scaffold was similar to PDA/RGD-SG/PCL and PDA/RGD-MG/PCL scaffolds. However, 1% PDA/RGD-graphite/PCL scaffold displayed much worse electric conductivity than 1% PDA/RGD-SG/PCL and 1% PDA/RGD-MG/PCL scaffolds (Peer review file figure 2). Similarly, we further compared cell viability among 1% single-layered graphene/PCL, 1% multi-layered graphene/PCL and 1% graphite/PCL scaffold via CCK assay. The results showed the best cell viability from 1% single-layered graphene/PCL, followed by multi-layered graphene/PCL. Graphite/PCL scaffold showed the worst outcome. We reached similar results from LIVE/DEAD cell staining and Ki67 immunofluorescent staining (Peer review file figure 1 and 3). In addition, phalloidin staining showed that cell density was much lower on graphite nanoparticles based PCL scaffolds in comparison with SG and MG based scaffolds. The protuberances were more extended on graphene nanoparticles based PCL scaffolds than graphite based scaffolds (Peer review file figure 4). This further indicated the graphite based PCL scaffolds were not as good as graphene/PCL scaffolds in supporting cell attachment and proliferation.

These results validated the statement that under the same concentration, 1% SG and MG in PCL scaffolds can better improve cell viability than 1% graphite based PCL scaffolds, due to their electric conductivity, and micropatterning.

Apart from the cell viability, we also evaluated the potential effects of graphite based PCL scaffolds on RSCs proliferation, attachment and neural expression. The immunofluorescent staining of S100, Tuj1 and GFAP was performed respectively. The results showed that 1% SG and MG based PCL scaffolds could better improve

RSC neural expression than 1% graphite based scaffolds (Peer review file figures 5-7). We further performed Western blotting assay and reached similar conclusions (Peer review file figure 8). We evaluated the Brdu, Ki67, N-cadherin, vinculin, GFAP and Tuj1 expression of RSCs cultured on different scaffolds. The results from PDA/RGD-SG/PCL and PDA/RGD-MG/PCL scaffolds were better than the others.

In addition, we also figured out that apart from electric conductivity, the graphene based materials also contributed to peripheral nerve regeneration with their biochemical cues and morphological stimuli. The graphene material was confirmed to work as a neurogenesis inductive substrate. It could offer mechanophysical cues via exerting stretch stimulation on Schwann cells [Biochemical and Biophysical Research Communications 2015; 460: 267-273]. Our three-dimensional PDA and RGD coated single-layered and multi-layered graphene poly-caprolactone scaffolds are able to provide ideal cell signals including chemical signals from RGD and PDA coating and electrical signals from graphene conductivity of bioelectricity. Previously, Egeland et al. reported that they used poly(3,4-ethylenedioxythiophene)-modified neural interfaces to conduct action potential and stimulate biophysiological reaction through a significant nerve defect. Besides, the free electrons from the environment creates certain current through graphene material [Adv. Mater. 2011; 23: H263–H267.]. Moreover, the macroporous and nanoporous structures equip 3D graphene based materials with special topological characteristics for cell growth and attachment because they offer free entrance of water, protein and other nutrition molecules. Therefore, an appropriate combination of morphological and bio-electrically conductive stimuli was used in the fabrication of our graphene based materials. Thank you very much for your advices!

6. In line 132-133 the authors claimed that the electrical conductivity than other conductive materials, this may not be true.

RE: We revised it according to your advices. Thank you for your important opinion! In this study, we fabricated graphene based nanomaterials to promote peripheral nerve regeneration. These materials were equipped with some merits including ideal biocompatibility, bioelectric conductivity and morphological stimuli. These are important for a successful nerve conduit fabrication. In previous studies, Masand et al. fabricated a peptide modified nerve conduit with polysialic acid and human natural killer cell epitope and found it was very effective for myelination, axonal and motor neuron recovery in a mouse femoral nerve defect model. Huang et al. used active silk conduit in rat sciatic nerve repair and discovered that long gaps like 11 and 13 mm could be successfully repaired after 12 weeks of injury [Biomaterials 2012; 33: 8353-8362.]. Wu et al. fabricated a bioactive polyurethane nanoscaffold and discovered that it could upregulate neurotrophin expression by activating voltage gated calcium channel to improve peripheral nerve regrowth [Biomaterials 2012; 33:59-71]. Wang et al. fabricated nerve scaffolds with different stiffness and evaluated their roles in peripheral nerve regeneration. In their

research, higher PCL concentration in the poly(propylene fumarate)- co -polycaprolactone scaffold better improved peripheral nerve function and structural reconstruction [Biomaterials 2016; 87:18-31.]. Polypyrrole (PPY) is an excellent conductive material [Adv. Funct. Mater. 2015; 25: 2715-2724.]. Our research team previously evaluated the conductivity of PPY based PLCL conduit and the value was 6.72×10^{-5} S/cm [Front Mol Neurosci. 2016; 9: 117.]. It is true that our graphene materials did not exhibit as good conductivity as some other metallic materials. However, the low concentration of graphene is more appropriate for nerve regeneration. Thank you very much for your advices!

7. In line 287 the authors claimed that “This assured the balanced and strong electrical conductivity for peripheral nerve regrowth.” I wonder if the authors like to say the balanced conductivity is better or the higher the better. It is not clear.

RE: We are sorry for this inaccurate expression. Actually, we tend to express that it assured the ideal electric conductivity for peripheral nerve regrowth. It is not exactly the stronger the conductivity, the better nerve recovery is. Appropriate electrical signaling is one of the vital factors to nerve functional restoration. Therefore, we have revised it in our updated manuscript accordingly. Besides, in our fabrication of graphene based nanomaterials, we considered the significance of biocompatibility and cytotoxicity of the materials in peripheral nerve regeneration. Our three-dimensional PDA and RGD coated single-layered and multi-layered graphene poly-caprolactone scaffolds are able to provide ideal cell signals including chemical signals from RGD and PDA coating and electrical signals from graphene conductivity of bioelectricity. Previously, Egeland et al. reported that they used poly(3,4-ethylenedioxythiophene)-modified neural interfaces to conduct action potential and stimulate biophysiological reaction through a significant nerve defect [Plast Reconstr Surg. 2010; 126: 1865-1873.]. Besides, the free electrons from the environment creates certain current through graphene material [Adv. Mater. 2011; 23: H263–H267.]. In addition, the macroporous and nanoporous structures equip 3D graphene based materials with special topological characteristics for cell growth and attachment because they offer free entrance of water, protein and other nutrition molecules. An appropriate combination of morphological and bio-electrically conductive stimuli was used in the fabrication of our graphene based materials. Thank you very much for your advices!

8. The printing skill is just a layer-by-layer printing, it is not necessary to use a 3D printer which is more complicated.

RE: Thank you so much for your opinion! It is true that layer-by-layer (LBL) printing is very efficient in biomaterial fabrication. However, in our study, the 3D printing enabled us to fabricate the conduit with a bottom-up style, which means layer by layer. Fluid droplets are sprayed onto the printing surface as a dot-to-dot style followed by a previously introduced pattern [Cell Biochem Biophys. 2016; 74: 93–98.]. Besides, the printer also allowed digital control of mixed solution injection, and this contributed to an even distribution of different materials in the conduit.

Also, the 3D printing allowed us to create a conduit with certain volume of different elements and fabricate it from any angle, position, or plane. Besides, with high resolution, 3D printing could better improve RSC proliferation, attachment and neural expression. In order to validate our statement, we performed some immunofluorescent assays. We fabricated graphene/PCL conduits via layer by layer method only, and compared its effects with 3D printing graphene/PCL scaffolds. The immunofluorescent assays of Ki67, phalloidin, S100, GFAP and Tuj1 all indicated that graphene based PCL conduits via 3D printing and LBL method showed better results than the conduits made via LBL method only (Supplementary Peer review file figures 9-14). From these results, we can know that 3D printing and LBL fabrication method is more appropriate for successful peripheral nerve regeneration in this study.

Peer Review File Figure 10. Phalloidin staining of RSC on different nanoscaffolds. (a-c) Phalloidin staining on PDA/RGD-SG/PCL(LBL). (d-f) Phalloidin staining on PDA/RGD-MG/PCL(LBL). (g) Cell density evaluation from phalloidin staining. The scale bar is 50 μm. All data are displayed as mean ± standard deviation. #p<0.05 compared with PDA/RGD-PCL; ^Δp<0.05 compared with PCL; ^θp<0.05 compared with PDA/RGD-SG/PCL(LBL); ^φp<0.05 compared with PDA/RGD-MG/PCL(LBL) (the statistical test is ANOVA).

Peer Review File Figure 11. Immunofluorescent staining for Ki67. (a-c) Ki67 expression of SC on PDA/RGD-SG/PCL(LBL). (d-f) Ki67 expression of SC on PDA/RGD-MG/PCL(LBL). (g) Relative expression of Ki67. The scale bar is 50 μm. All data are displayed as mean ± standard deviation. * $p < 0.05$ compared with PDA/RGD-MG/PCL; # $p < 0.05$ compared with PDA/RGD-PCL; $\Delta p < 0.05$ compared with PCL; $\phi p < 0.05$ compared with PDA/RGD-SG/PCL(LBL); $\Phi p < 0.05$ compared with PDA/RGD-MG/PCL(LBL) (the statistical test is ANOVA).

Peer Review File Figure 12. Immunofluorescent staining for S100. (a-c) S100 expression of SC on PDA/RGD-SG/PCL(LBL). (d-f) S100 expression of SC on PDA/RGD-MG/PCL(LBL). (g) Relative expression of S100. The scale bar is 50 μm. All data are displayed as mean ± standard deviation. #p<0.05 compared with PDA/RGD-PCL; Δp<0.05 compared with PCL; θp<0.05 compared with PDA/RGD-SG/PCL(LBL); φp<0.05 compared with PDA/RGD-MG/PCL(LBL) (the statistical test is ANOVA).

Peer Review File Figure 13. Immunofluorescent staining for GFAP. (a-c) GFAP expression of SC on PDA/RGD-SG/PCL(LBL). (d-f) GFAP expression of SC on PDA/RGD-MG/PCL(LBL). (g) Relative expression of GFAP. The scale bar is 50 μ m. All data are displayed as mean \pm standard deviation. * $p < 0.05$ compared with PDA/RGD-MG/PCL; # $p < 0.05$ compared with PDA/RGD-PCL; $\Delta p < 0.05$ compared with PCL; $\rho p < 0.05$ compared with PDA/RGD-SG/PCL(LBL); $\phi p < 0.05$ compared with PDA/RGD-MG/PCL(LBL) (the statistical test is ANOVA).

Peer Review File Figure 14. Immunofluorescent staining for Tuj1. (a-c) Tuj1 expression of SC on PDA/RGD-SG/PCL(LBL). (d-f) Tuj1 expression of SC on PDA/RGD-MG/PCL(LBL). (g) Relative expression of Tuj1. The scale bar is 50 μ m. All data are displayed as mean \pm standard deviation. #p<0.05 compared with PDA/RGD-PCL; Δ p<0.05 compared with PCL; Φ p<0.05 compared with PDA/RGD-SG/PCL(LBL); Φ p<0.05 compared with PDA/RGD-MG/PCL(LBL) (the statistical test is ANOVA).

In addition, it could also be mimic *in vivo* structure and environment of tissues by 3D scaffolds fabrication. In our study, we added PDA and RGD to enhance cell attachment and viability in the conduit with 3D printing method. Moreover, 3D guided printing method significantly decreased the overall time spent on fabrication. In this way, 3D printing is important for conduit fabrication.

9. Regarding the description in line 347, the polarity of pi-pi bonds in graphene should be very weak.

RE: Yes. π - π bonds in graphene based nanomaterials could lead to good biocompatibility, nerve cell proliferation, attachment and differentiation. π - π bonds enabled interaction between graphene based materials and other DNA, peptides and molecules. In our study, we aimed to fabricate a biocompatible graphene nanoscaffold with PDA and RGD coating and recreate an ideal microenvironment around injured peripheral nerves [Adv Funct Mater. 2008; 18: 3506-3514].

Reviewer #3 (Remarks to the Author):

The authors report 3D-printing of a novel graphene based nerve conduit to improve peripheral nerve regeneration. While polydopamine (PDA) and arginylglycylaspartic acid (RGD) have been used previously for coating nerve guidance channels/scaffolds/conduits, both compounds have been combined in this study for the first time for coating of conduits to protect against cytotoxicity of single-layered and multi-layered graphene. Moreover, long-term *in vivo* studies to improve peripheral nerve recovery after injury using graphene based conduits with or without Schwann cell seeding have not been performed previously. The authors report that the overall outcome of the Schwann cell loaded PDA/RGD-SG/PCL and PDA/RGD-MG/PCL resembles the outcome achieved with an autograft. This reported finding is remarkable since, to my opinion, no artificial scaffold was able to match the regenerative improvement seen with a peripheral nerve autograft with respect to axon elongation, myelination and functional outcome. However, there are a number of questions and specific concerns to be raised.

Fig. 3E,3F Cytotoxicity assay: What are the units of measure at the Y-axis?

RE: Thank you for your opinion! The units of measure at the Y-axis are OD. We conducted cytotoxicity assay via CCK 8. It is a basic and important agent used for cell proliferation and cytotoxicity. In our study, we evaluated the optic density (OD) results in CCK 8 assay measured by microplate reader. The higher the values are, the better the cell proliferation condition is. We felt sorry for this error. It was corrected in the revised figures.

Fig. 10, F-6: the staining of (myelinated) axons is very weak, suggesting that the number of axons in the autograft is lower than in graphene conduits (F-1 and F-2). This is in contrast to the histogram in Fig. 9H and text line 261/262 that reads: "The number of myelinated axons was significantly higher in autograft group...".

Moreover, the staining used in this figure (HE, Toluidine Blue) to indicate myelinating axons is questionable. A specific antibody to stain (peripheral) myelinated fibers should be used, e.g. antibodies directed to P0 or MBP.

RE: We are sincerely sorry for this mistake. Actually the staining of myelinated axons from F-6 Fig.8 cannot represent a satisfactory results of autograft transplantation. We have replaced it with other pictures from autograft group. As for the staining used in this set, we tended to evaluate axonal regrowth and overall area from the examination. The TEM observation is more appropriate for myelination evaluation. Besides, it is very important for us to add a specific marker to represent the exact recovery of myelinated fibers. The staining of MBP was added in the revised manuscript, along with a specific Schwann cell marker S100 as triple immunofluorescent staining from nerve samples at 6, 12 and 18 weeks postoperatively respectively (Fig. 8, Supplementary Figs. 12 and 13). Thank you for your advices!

Fig. 11 (TEM): The TEM resolution is very poor. (a) The layers of the myelin sheath are not visible and (b) no basement membrane, a structural characteristic of Schwann cells,

can not be seen.

RE: We are sincerely sorry again for the unsatisfactory resolution of TEM pictures. It probably results from too many pictures in one figure. We have made relevant changes to reorganize these pictures and better displayed the characteristics of myelin sheath, Schwann cell structure in the revised manuscript and figure sets (Fig. 8, Supplementary Figs. 7 and 8). Thank you for your advices!

Fig. 12: This figure is incomplete. Panels A6-R6 are missing.

RE: We are sorry for bringing you the inconvenience and misunderstanding. This set of pictures are representatives for NF200 and β -tubulin triple immunofluorescent staining. Each line represented one group at certain time nodes. For instance, Line 1 represented Schwann cell loaded single-layered graphene/PCL conduit group samples. From left to right, the four pictures are nucleus staining, marker 1 staining, marker 2 staining and merged picture. It is the same to line 2, which is Schwann cell loaded multi-layered graphene/PCL conduit group samples. There are no A6 to R6 or A5 to R5 in this figure. Please refer to the revised manuscript for these figures. Thank you for your advices!

While in line 218 "walking track analysis" is mentioned to evaluate functional recovery, no evidence besides measuring toe distance variables is provided for locomotor/walking improvement.

RE: Thank you for your suggestion! We also noticed that apart from SFI, another important factor could be taken into consideration for walking improvement. Extensor postural thrust is an important indicator measured at the same time with walking track analysis to evaluate the overall locomotor improvement. It is characteristic of postural reflex reaction and therefore it is associated with SFI. In brief, the injured leg was in contact with metatarsus for 30 seconds, and the largest pushing force was recorded. The experiment was repeated for five times. From the results, the Schwann cell loaded single-layered and multi-layered graphene/PCL conduit generally exhibited a better pushing force than non-cell loading conduit group at 6, 12, and 18 weeks postoperatively. Meanwhile, they were similarly excellent in comparison with autograft group at week 18 (Supplementary Fig. 4). This further indicated the beneficial effects of graphene based nerve conduit and cell loading in the sciatic nerve functional recovery. Thank you for your advices!

Line 271: "Tuj1 stood for migration of Schwann cells". Tuj1/ β -III tubulin is an axonal marker in peripheral nerve rather than a Schwann cell marker. Tuj1 staining of axons could explain why the two histograms in Fig. 12S and 12T look identical. One would expect this since both NF200 and Tuj1 are axonal markers. To specifically label Schwann cells an antibody directed to S100 β should be used.

RE: Thank you for your valuable suggestion! We have also noticed similar NF200 and Tuj1 staining results in representing axonal regrowth. Therefore, the graphs between these two markers resembled with each other in Fig 12S and 12T. We would be honored to accept your advice and add the S100 staining in the revised

manuscript. As a result, we performed the MBP/S100 triple immunofluorescent staining to better address these issues (Fig. 10, Supplementary Figs. 10 and 11). Thank you for your advices!

Line 290: the statement "...first time that biodegradable materials have been used with single-layered and multi-layered graphene..." is not proven as no biodegradability has been shown in this study - at least not within one year.

RE: Thank you for your valuable opinion! We would like to clarify our intentions underlying this statement. As for the biodegradation of graphene based PCL conduit *in vivo*, we did not notice full degradation at 18 weeks postoperatively but only felt the conduit was much softer than it was at implantation. From previous articles, the substrate material PCL generally degrade fully within 6-12 months *in vivo* [J Macromol Sci Pure Appl Chem. 1994; 32: 867-873.]. Besides, it is biocompatible and poses no toxicity to the living body. We would expect that with the present size and concentration of PCL in the conduit, the nerve conduit is very likely to experience full degradation after a relatively long period of time. Based on above considerations, we made the conclusion that it is the first time that biodegradable materials have been used with single-layered and multi-layered graphene for nerve conduit fabrication. Thank you for your advices!

Line 294/295: "These qualities make it the best scaffold material for ideal nerve function restoration." This statement regarding Schwann cell seeded graphene based conduits is not valid since no direct comparison with other scaffolds is shown.

RE: We feel sincerely sorry for this inaccurate statement. From our *in vivo* experiments, we find the Schwann cell seeded graphene based conduits had huge potentials in restoring functional and morphological recovery of sciatic nerves. Actually, we tend to compliment it as an ideal nerve conduit in this field. We have corrected this statement in the revised manuscript. Thank you for your advices!

Line 309/310: "Thus, extra electrical signals are needed to replace the damaged signaling transduction system for efficient nerve restoration". This statement indicates that electrical activity of the conduit could replace the interrupted signaling transduction in the injured nerve. This is very unlikely. To my opinion, there is no prove in the literature for this assumption.

RE: We are sorry for this inaccurate assumption. As a matter of fact, we tended to express our opinions on the excellent electrical signal transduction of graphene based conduit and its important role in peripheral nerve regeneration. However, the word 'replace' is highly inappropriate in the context. Moreover, the graphene based materials were very important for peripheral nerve regeneration due to its bioelectrical property and its morphological characteristics. The macroporous and nanoporous structures equip 3D graphene based materials with special topological characteristics for cell growth and attachment because they offer free entrance of water, protein and other nutrition molecules. An appropriate combination of morphological and bio-electrically conductive stimuli was used in the fabrication of

our graphene based materials. Therefore, we have deleted this sentence in the revised manuscript. Thank you for your advices!

Discussion line 372/373: "The graphene based nanomaterials bring hope and light to people for curing long-range nerve defexcts in the near future". This outlook is overstretching the data and misleading. As the axon bridging distance in the present paper was in the same range (1.5 cm) as in many other successful peripheral nerve regeneration studies using conduits. Whether or not graphene based conduits do better in long distance bridging remains to be seen. There is no evidence in the present paper for bridging longer gaps between nerve stumps than 1.5 cm.

RE: Thank you for your valuable opinions! We revised it according to your advices. Researchers used different lengthy nerve defects in their experiments. The most commonly adopted length was 1 cm because most materials could help the peripheral nerve regeneration within this range. Long range defects for 1.5 cm or above have been attempted in some articles successfully. According to our observation, most researchers evaluated *in vivo* experiments at three months after surgery. In our study, we evaluated all the results from *in vivo* assays after 18 weeks (approximately 4 and a half months). Moreover, with our evaluation, the outcomes from Schwann cell loading single-layered graphene and multi-layered graphene nerve conduit exhibited much better performances in functional, electrophysiological and morphological nerve recovery than non-cell-seeding conduit groups. And they resembled the outcomes from autograft group at week 18 postoperatively. This is an ideal result that we all expect from an artificial nerve conduit in the peripheral nerve regeneration. Therefore, we expect its future roles in the long-range nerve defect repair. However, its application in human remains further investigation. Thank you for your advices!

An important question remains: Is there evidence for sensory recovery following grafting of graphene loaded nanoscaffolds?

RE: Thank you for your significant suggestion! It is a very important issue that we must address in the revised manuscript. We added thermal sensitivity experiment followed by a previous article [Hand Surg Am. 2015; 40: 314-322.]. From our results, all the rats in conduit implantation groups exhibited longer reaction time than autograft at 6 and 12 weeks. However, at 18 weeks, the cell loading conduit groups showed similar response time in comparison with autograft and the results were significantly better than the rest groups (Supplementary Fig. 4). This indicated a positive effect in sensory recovery for sciatic nerves under graphene based conduit application and cell loading therapy. The details could be referred to in our revised manuscript and figures. Thank you for your advices!

Finally, there are many typing errors in the text.

RE: We feel sorry for these mistakes. We have asked language experts for help to improve the grammar and phrase use in our revised manuscript carefully. Thank you for your valuable opinion!